

# BrGDGT-based palaeothermometer in drylands: the necessity to constrain aridity and salinity as confounding factors to ensure the robustness of calibrations.

Lucas Dugerdil[1,2], Sébastien Joannin[1], Odile Peyron[1], Shafag Bayramova[3], Xiaozhong Huang[4,5], Fahu Chen[4,6,7], Dilfuza Egamberdieva[8,9], Jakhongir Alimov[10,11], Bazartseren Boldgiv[12], Amy Cromartie[13,1], Juzhi Hou[6,7], Lilit Sahakyan[14], Khachatur Meliksetian[14], Salomé Ansanay-Alex[2], Rafig Safarov[15], Imran Muradi[15], Shabnam Isayeva[16], Shehla Mirzayeva[16], Elshan Abdullayev[17,18], Sayyara Ibadullayeva[16], Parvana Garakhani[16], and Guillemette Ménot[2]

[1]ISEM, Univ Montpellier, CNRS, IRD, Montpellier, France
[2]ENS Lyon, UCBL, CNRS, UMR 5276 LGL-TPE, Lyon, France
[3]Institute of Geology and Geophysics, Department of modern geodynamics and space geodesy, Azerbaijan
[4]Key Laboratory of Western China's Environmental Systems (Ministry of Education), College of Earth and Environmental Sciences, Lanzhou University, Lanzhou, China
[5]Yunnan Key Laboratory of Plateau Geographical Processes and Environmental Changes, Faculty of Geography, Yunnan Normal University, Kunming 650500, China
[6]Alpine Paleoecology and Human Adaptation Group, State Key Laboratory of Tibetan Plateau Earth System, Resources and Environment, Institute of Tibetan Plateau Research, Chinese Academy of Sciences, Beijing, China
[7]College of Resources and Environment, University of Chinese Academy of Sciences, Beijing, China
[8]Institute of Fundamental and Applied Research, National Research Univ. (TIIAME), Tashkent 100000, Uzbekistan
[9]Leibniz Centre for Agricultural Landscape Research (ZALF), 15374 Müncheberg, Germany
[10]International Agriculture University, Kibray district, 100164, Tashkent, Uzbekistan
[11]Faculty of Biology, National University of Uzbekistan, 100174, Tashkent, Uzbekistan
[12]Ecology Group, Department of Biology, School of Arts and Sciences, National University of Mongolia, Ulaanbaatar 14201, Mongolia
[13]Université Côte d'Azur, CNRS, CEPAM, UMR 7264, 06300, Nice, France
[14]National Academy of Sciences, Institute of Geological Sciences of Armenia, ave. M. Baghramyan 24a, Yerevan 0019, Armenia
[15]Institute of Geology and Geophysics, Science and Education Ministry of Azerbaijan Republic, AZ1073, Baku, Azerbaijan
[16]Institute of Botany, Azerbaijan National Academy of Sciences, Baku, Azerbaijan
[17]Institute of Geography and Geology, University of Greifswald, Friedrich-Ludwig-Jahn Strasse 17A, D-17487 Greifswald, Germany
[18]Department of Life Sciences, Khazar University, 41 Mahsati Str., AZ1096 Baku, Azerbaijan

**Correspondence:** Lucas Dugerdil (lucas.dugerdil@ens-lyon.fr) and Guillemette Ménot (guillemette.menot@ens-lyon.fr)

**Abstract.** Past temperature reconstructions offer valuable insights into the impact of climate change on the global climate-human-vegetation system. Branched glycerol dialkyl glycerol tetraethers (brGDGTs) are recognized as effective temperature proxies, particularly in lakes and peatlands, where they are well preserved. However, their reliability as palaeothermometers can be compromised by factors beyond air temperature, especially in drylands. This study introduces the Arid Central Asian (ACA) brGDGT surface Data Base, a regional dataset consisting of 162 new surface samples from the drylands of ACA, in addition to 599 previously published samples. The distribution of brGDGTs in relation to climate and environmental variables



was analysed to explore their potential as reliable temperature proxies, mainly focusing on brGDGTs methylation (MBT), cyclisation (CBT), and isomer (IR) indices. The brGDGT-based palaeothermometer is a promising tool for understanding past climates, but our comparison between an ACA-centred database and a worldwide continental surface sample database reveals several challenges. Drylands exhibit extreme climate and soil/lacustrine properties, amplifying the impact of confounding factors on brGDGT-based relationships with mean annual air temperature. Salinity emerges as the dominant factor influencing brGDGT variance, followed by sample type, salinity, pH, and aridity, all of which contribute significantly. These factors interact in complex ways, with the salinity effect varying between soil and lacustrine deposits. For sample physicochemical conditions, the $\mathrm{IR}'_{6+7\mathrm{Me}}$ index is best for salinity, and $\mathrm{IR}_{6\mathrm{Me}}$ is most suitable for pH reconstruction. Despite this, the $\mathrm{MBT}'_{5\mathrm{Me}}$-temperature relationship is limited in ACA, particularly for lacustrine samples, and $\mathrm{MBT}'_{6\mathrm{Me}}$ does not offer a better solution under hyper- to semi-arid conditions. Sub-calibrating models for specific environmental conditions such as salinity and aridity improves the accuracy of temperature reconstructions. Furthermore, the difference between $\mathrm{MBT}'_{5\mathrm{Me}}$ and $\mathrm{MBT}'_{6\mathrm{Me}}$ provides a promising proxy to assess aridity. Although the brGDGT signal in drylands is influenced by multiple confounding factors, it remains a valuable tool for understanding past climate and environmental conditions, especially when accounting for the complex interactions between these factors based on each study's unique physicochemical and bioclimatic context. Further research, incorporating a broader range of surface samples alongside comprehensive soil and climate data, holds the potential to enhance the accuracy of brGDGT-based climate reconstructions.

## 1 Introduction

Given the uncertain implications of the anthropogenic climate change on the environment, hydrology and human society, reconstructions of the past climate temperatures provide a comprehensive perspective on the impact of climate change on the climate-human-vegetation system (Tierney et al., 2020). Branched glycerol dialkyl glycerol tetraethers (brGDGT) are new promising temperature proxies that have been used on continental archives (Weijers et al., 2007; Peterse et al., 2012; De Jonge et al., 2014; Dearing Crampton-Flood et al., 2020; Raberg et al., 2021), especially since these lipid compounds produced from bacterial membranes are ubiquitous (Raberg et al., 2022b), well preserved on past archives of lakes (Dang et al., 2016b; Wang et al., 2021; So et al., 2023) and peatlands (Naafs et al., 2017a, b; d'Oliveira et al., 2023). From laboratory experiments (Halamka et al., 2023), simulations (Naafs et al., 2021), but especially from soil surface samples (De Jonge et al., 2014; Dearing Crampton-Flood et al., 2020) or lake surface sediments (Sun et al., 2011), the number of methyl groups on the aliphatic chain of brGDGT changes with air temperature, as shown by Weijers et al. (2007) and De Jonge et al. (2014). The relationship is clear and linear. The diverse applications of brGDGT-based palaeothermometers cover a wide range of environments and archives: from tropical (Pérez-Angel et al., 2020; Häggi et al., 2023) to arctic lakes (Raberg et al., 2022a); from acid (Dang et al., 2016a) to alkaline lakes (Yang et al., 2014; Dang et al., 2016a); and also from loess-palaeosol sequences (Lin et al., 2024), freshwater (Dugerdil et al., 2021b; Robles et al., 2022) to saline lakes (Wang et al., 2021; So et al., 2023). However, the change in the relative proportion of brGDGTs does not depend solely on the air temperature, which can significantly undermine the reliability of the palaeothermometer.



The influence of pH on brGDGT distribution was initially identified and thoroughly constrained (Weijers et al., 2007). pH primarily influences the relative number of 5- and 6-methyl isomers and the number of cyclisations along the aliphatic chain, while temperature affects the number of methylations (e.g., higher amount of tetramethylated compounds over warm environment, Sun et al., 2011; Peterse et al., 2012; Dang et al., 2016a; Raberg et al., 2022b). De Jonge et al. (2014) present the application of the index of methylation of branched tetraethers ($\mathrm{MBT}'_{5Me}$) index associated with mean annual air temperature

(MAAT), the isomer ratio (IR), and the cyclisation of branched tetraethers ($\mathrm{CBT}'$) indices related to pH. These two important indices are now widely adopted to calibrate the reconstruction of pH and MAAT in the past by linear relationships. The sample type, including peat, soil, river, marine, loess, and lacustrine samples, influences the brGDGT-temperature relationship (Loomis et al., 2011; Martínez-Sosa et al., 2023). At the global scale, various specific calibrations between $\mathrm{MBT}'_{5Me}$ and MAAT have been suggested for soil (De Jonge et al., 2014; Chen et al., 2021), peat (Naafs et al., 2017b), and lacustrine samples (Sun et al.,

2011; Zhao et al., 2021). The influence of the calibration database size and the biogeographical characteristics of brGDGT distribution has also been examined (Dugerdil et al., 2021a), and several local or regional calibrations allow for more accurate MAAT reconstructions in past archives (Chen et al., 2021). Subsequently, an increased number of confounding factors have been identified, e.g., soil moisture (Dang et al., 2016b) and sample type (e.g., soil or lacustrine; Martin et al., 2019; Martínez-Sosa et al., 2021; Martínez-Sosa et al., 2023); temperature seasonality (Deng et al., 2016; Dearing Crampton-Flood et al.,

2020), or vegetation (Häggi et al., 2023). Thus, the lacustrine samples have lower abundances of pentamethyls than soil samples (De Jonge et al., 2014; Martin et al., 2019; Raberg et al., 2022b), while peat samples are dominated by tetramethyls (Naafs et al., 2017a). The relationship between brGDGTs and MAAT may exhibit a bias toward the summer temperatures, particularly in soils and lakes that experience a long frost period (Deng et al., 2016; Dearing Crampton-Flood et al., 2020). Vegetation influences the distribution of brGDGTs, likely due to the higher soil organic content found in vegetated soils compared to bare

soils (Liang et al., 2019). This leads to a differential relationship between brGDGTs and MAAT across different vegetation communities (Häggi et al., 2023).

   Confounding factors introduce various biases depending on geographic and climatic contexts. While they have been extensively studied in tropical regions and high-latitude or high-altitude environments (Pérez-Angel et al., 2020; Raberg et al., 2021; Zhao et al., 2021; Häggi et al., 2023), they remain poorly constrained in semi-arid to hyper-arid areas (Yang et al., 2014;

Duan et al., 2020; Guo et al., 2021). In drylands, limited and erratic precipitation is the primary water input, critically influencing soil moisture. This persistent water deficit intensifies aridity, which is defined by the imbalance between precipitation and evapotranspiration (Trabucco and Zomer, 2018). Such bioclimatic stress affects soil chemistry, often reducing leaching and causing the accumulation of base cations (e.g., calcium, magnesium), which contributes to alkaline soils with low organic matter (Muhammad et al., 2008). Coarse-textured, well-drained soils are also common, increasing the occurrence of gypsum

or saline profiles (Plaza et al., 2018). Additionally, sparse vegetation - further stressed by intense grazing pressure - exacerbates land degradation (Maestre et al., 2022). The combined effects of aridity and overgrazing increase soil vulnerability to erosion, reinforcing a cycle of organic matter depletion, nutrient loss, and alkaline soil dominance (Moreno-Jiménez et al., 2019). As a result, brGDGT-based reconstructions in drylands are especially prone to biases driven by bare soil conditions, aridity, and soil chemistry impacts on bacterial communities.





**Figure 1.** Distribution map of the global surface samples presented in this study (WDB, **A**) followed by the extent of the Arid Central Asian brGDGT surface Data Base (ACADB, **B**) and local focus on (**C**) Qilian Shan and (**D**) Caucasus surface samples. (**E**) distribution of brGDGT sampling sites in the ACA bioclimate space. The elevation map comes from SRTM Digital Elevation Model version 4.1 (Jarvis et al., 2008), the Aridity Index from CGIAR (Trabucco and Zomer, 2018) and the extracted climate parameters from worldclim2.1 (Fick and Hijmans, 2017) with mean air temperature of Months Above Freezing, Mean Annual Air Temperature, Mean Annual Precipitation, Mean Temperature/Precipitation of the Cold and Warm Quarters.

Primarily arid soils, characterized as soils receiving less than 500 mm.yr$^{-1}$ (Peterse et al., 2012; Yang et al., 2014; Guo et al., 2021), present higher brGDGT variability and specific isomer distribution (Guo et al., 2021). The comparison of soil samples from drylands indicates a dissimilarity to global brGDGT databases (Yang et al., 2014; Dearing Crampton-Flood et al.,





2020; Véquaud et al., 2022). The main difference is the higher frequency of 6- compared to 5-methyl compounds (Duan et al., 2020). This distinct methylation process may explain the reduced statistical strength for brGDGT-temperature calibration under

arid conditions (Peterse et al., 2012; Wang et al., 2019). Moreover, the temperature control over tetramethylated compounds may be related to 5- or 6-methyl assemblages in diverse contexts, likely due to specific bacterial communities, mitigating the reliability of $\mathrm{MBT}'_{5\mathrm{Me}}$-based temperature reconstruction (Wang et al., 2024). Additionally, Duan et al. (2022) report the influence of pH and, in particular, alkalinity on the distribution of brGDGTs in dry soils. A few recent studies have reported the impact of salinity on brGDGTs (Wang et al., 2021; Raberg et al., 2021; Kou et al., 2022; So et al., 2023). Salinity is

expected to influence the relative number of 5-, 6- and 7-methyl isomers (Wang et al., 2021). This effect impacts the $\mathrm{MBT}'_{5\mathrm{Me}}$- and $\mathrm{MBT}'_{6\mathrm{Me}}$-based temperature reconstructions (Kou et al., 2022; So et al., 2023). Although several palaeosalinity proxies have been proposed to address these biases, significant work remains to be undertaken, for instance, on the precise ionic composition of soil (Chen et al., 2022; De Jonge et al., 2024a). Among the possible solutions to reduce these biases, the brGDGT-temperature relationship independent of these two indexes includes Multiple linear Regressions (MR; De Jonge et al.,

2014; Raberg et al., 2021), Bayesian calibrations (Dearing Crampton-Flood et al., 2020), and machine learning approaches (Véquaud et al., 2022). However, the coupled effects of aridity, pH and salinity on soil, loess, and lacustrine archives can significantly alter the interpretations of brGDGT-based climate reconstructions in the ACA region (Lin et al., 2024).

This study presents the first regional database of surface brGDGT samples for drylands, aiming to identify the key climate and environmental parameters influencing their distribution. This dataset, referred to as the Arid Central Asian brGDGT Surface

Database (ACADB), includes brGDGT assemblages from various sites across the region, totalling 761 sites. It includes 162 new sites collected from four countries in ACA and aggregated with 599 previously published samples from ACA (Fig. 1). These modern samples are analysed in relation to key climate parameters, mainly aridity, temperature, and precipitation, as well as chemical characteristics such as pH, salinity, and sample type (soil or lacustrine). The results are then compared with the global Worldwide brGDGT Surface Database (WDB; modified from Raberg et al., 2022a) to assess whether similar brGDGT

patterns are observed at both regional and global scales. The methodological approach is synthesized in Fig. 2. Following this workflow, this study raises the following questions:

1. Which confounding factor - pH, aridity, salinity, or sample type - has the most significant impact on brGDGT-temperature calibrations in drylands?

2. Are the $\mathrm{MBT}'_{5\mathrm{Me}}$ and $\mathrm{MBT}'_{6\mathrm{Me}}$ reliable for reconstructing past temperature in this context?

3. Can we apply aridity- or salinity-related indices to soil and lacustrine samples similarly, or do we need to develop new indices or calibrations to track these confounding factors?





## 2  Materials and Methods

### 2.1  Study sites

This study pools new samples from four different ACA countries: Azerbaijan, China (from the Qaidam Basin to the Qilian
Mountains), Tajikistan, and Uzbekistan for a total of 162 samples (Fig. 2, step 1). No data has previously been published for
Azerbaijan and Uzbekistan. All site coordinates and geographic features are presented in Fig. 1 and in Table S.1. From the
Caucasus, 48 new surface sites are presented from Azerbaijan. The data location was randomly selected and covers all of
Azerbaijan, from the Great Caucasus to the Hyrcanian forest in the Talish Range, through the Lesser Caucasus Range, the Mil
Plain, and the Kura Valley (Fig. 1B). In China, an altitudinal/latitudinal gradient from the Qinghai Plateau to the southern part
of the Gobi Desert, through the Qilian Shan Range, presents 48 new surface samples (Fig. 1C). For the Tajikistan-Uzbekistan
database (TUSDB), the 66 sites come from the Aral Sea basin to the high Pamir-Alai Range. The site location and climatic
presentation of the TUSDB is also available in Dugerdil et al. (2025). The summarized information of each dataset is gathered
in Table 1.

   MAAT for the ACADB has an average value of 3.7 ± 3.2 °C and it is balanced between warm/mild environments (MAAT
> 10 °C), mainly on the western part of the ACA covering Caucasus (Armenia, Azerbaijan) and Middle Asia (Uzbekistan and
Tajikistan), and colder continental environments (MAAT < 3 °C) located in Central Asia (i.e., the southern part of Siberia, the
Mongolian plateau, and the Tibetan-Qinghai plateau in China, Table 1). On the opposite, the MAF is more homogeneous with
low MAF in China (MAF = 7.5 ± 2.7 °C) and high MAF in Tajikistan and Uzbekistan (13 ± 2.5 °C). This is due to the higher
seasonality in continental Central Asia than in the Caucasus and Middle Asia (Fig. S.1). Similarly, Mean Annual Precipitation is
spatially homogeneous, with consistent low values of $410 ± 140$ mm.yr$^{-1}$. However, the seasonal precipitation pattern varies
greatly across ACA. In the western region - including the Caucasus, Iran, and Middle Asia - winter dominates, with up to 65%
of annual rainfall occurring during this season. In contrast, the eastern region, encompassing the Central Asian plateaus and
southern Siberia, receives up to 87% of its precipitation in summer (Chen et al., 2024). This spatial diversity within the ACA
likely induces important heterogeneity in the bacterial growth season (i.e., fall and spring in western ACA, summer in eastern
ACA, Fig. S.1). In arid environments, brGDGT production may be influenced by water availability, potentially increasing
during the rainy season. This could bias reconstructed temperatures toward rainy-season conditions (De Jonge et al., 2014).
The ACA altitudinal gradient reached by the database covers -28 to 4038 m a.s.l. with an average value of about 1600 m a.s.l.

### 2.2  Environmental parameters

The new samples from this study (n = 162) are grouped into two main sample types: soil (n = 143) and lacustrine (n = 19)
samples. Here, *soil* means a sample collected on the surface from several subsamples collected within one m$^2$ area and from
the upper five cm part of the soil layer. *Lacustrine* corresponds to samples from lake sediment core-tops or surface sediments.
The majority of these lakes are in arid environments, seasonally dried, and associated with temporary ponds. For lacustrine
samples, one cm$^3$ of the upper parts of several top cores were sampled using a die-cut or from the upper five cm of the surface
sediment. For more details on the sample type description, please refer to Dugerdil et al. (2021a). For the sampling method



**Table 1.** BrGDGT dataset compiled in the Arid Central Asian Database (ACADB) and Worldwide Database (WDB) with their associated average climate parameters, data description (covered countries, dataset size, and site elevation), and original publications. New data published in this study are highlighted by a * for a total sum of 162 surface samples.

| Countries | N | Average bioclimate parameters[a] | | | | | Original publication |
| | | MAAT (°C) | MAF (°C) | MAP (mm.yr$^{-1}$) | AI | Altitude (m a.s.l.) | |
| --- | --- | --- | --- | --- | --- | --- | --- |
| ACA lakes | 52 | $1.8 \pm 3.4$ | $12 \pm 2.7$ | $260 \pm 120$ | $2500 \pm 1500$ | $1200 \pm 890$ | Wang et al. (2021) |
| Armenia | 22 | $6.8 \pm 3.4$ | $12 \pm 2.4$ | $420 \pm 110$ | $3600 \pm 1300$ | $1900 \pm 610$ | Cromartie et al. (2025) |
| Azerbaijan | **48*** | $12 \pm 3.3$ | $13 \pm 1.8$ | $510 \pm 160$ | $4200 \pm 1600$ | $560 \pm 630$ | This study |
| China | 120 | $3.7 \pm 6$ | $12 \pm 4.7$ | $290 \pm 170$ | $2800 \pm 2000$ | $1600 \pm 1300$ | Wang et al. (2020); Wang and Liu (2021); Zang et al. (2018); Dang et al. (2018) |
| China, Inner Mong. | 43 | $1.3 \pm 2.4$ | $12 \pm 1.1$ | $280 \pm 51$ | $2400 \pm 730$ | $930 \pm 310$ | Guo et al. (2021); Li et al. (2017) |
| China, Qinghai | **48*** | $-0.012 \pm 3.2$ | $7.5 \pm 2.7$ | $410 \pm 98$ | $4400 \pm 1900$ | $3200 \pm 680$ | This study |
| China, Tian Shan | 18 | $-2.1 \pm 4.6$ | $8.9 \pm 5.5$ | $280 \pm 76$ | $3900 \pm 2400$ | $2200 \pm 760$ | Duan et al. (2020) |
| China, Tibet | 129 | $-1.5 \pm 3$ | $6.1 \pm 1.4$ | $240 \pm 130$ | $2200 \pm 1200$ | $4600 \pm 240$ | Kou et al. (2022) |
| Global BayMBT Soils | 15 | $-1 \pm 2.5$ | $8.8 \pm 2.8$ | $330 \pm 110$ | $3000 \pm 1200$ | $2800 \pm 1800$ | Dearing Crampton-Flood et al. (2020) |
| Global soils | 48 | $0.46 \pm 3.6$ | $7.8 \pm 3.3$ | $380 \pm 76$ | $3400 \pm 1000$ | $3600 \pm 1500$ | Naafs et al. (2017a) |
| Mongolia | 31 | $1.1 \pm 2.4$ | $12 \pm 2.2$ | $230 \pm 78$ | $2200 \pm 1000$ | $1500 \pm 220$ | Dugerdil et al. (2021a) |
| Northern Iran | 48 | $17 \pm 0.84$ | $17 \pm 0.6$ | $330 \pm 15$ | $1900 \pm 110$ | $270 \pm 140$ | Duan et al. (2022) |
| Russia, Baikal | 20 | $-0.21 \pm 1.8$ | $9.3 \pm 0.9$ | $430 \pm 92$ | $5400 \pm 1000$ | $530 \pm 97$ | Khodzher et al. (2017); Dugerdil et al. (2021a); Wang et al. (2021) |
| Tajikistan | (53+**12***) | $4.7 \pm 4.6$ | $11 \pm 2.4$ | $470 \pm 260$ | $3700 \pm 1800$ | $2600 \pm 880$ | This study, Chen et al. (2021) |
| Uzbekistan | **54*** | $12 \pm 3.3$ | $14 \pm 2.2$ | $360 \pm 210$ | $2300 \pm 1600$ | $1000 \pm 760$ | This study |
| **Total ACA** | **761** | **$3.7 \pm 3.2$** | **$11 \pm 2.4$** | **$350 \pm 120$** | **$3200 \pm 1400$** | **$1900 \pm 720$** | |
| **World DB** | **2709** | **$12 \pm 8.8$** | **$14 \pm 6.6$** | **$1200 \pm 640$** | **$11000 \pm 6400$** | **$730 \pm 890$** | Raberg et al. (2022b)[b] |

[a] Bioclimate parameters refer to Mean Annual Air Temperature (MAAT), mean air temperature of Months Above Freezing (MAF), Mean Annual Precipitation (MAP), and Aridity Index (AI). Data were extracted from `worldclim2.1` (Fick and Hijmans, 2017) and `CGIAR` (Trabucco and Zomer, 2018). [b] for all original publications compiled in Raberg et al. (2022b), please report to Table S.2.

of the already published samples, refer to the original publications (Duan et al., 2020, 2022; Wang et al., 2021; Raberg et al., 2022b; Kou et al., 2022). In total, the ACADB is composed of 560 soil and 201 lacustrine samples.

The chemical characteristics of the ACA surface samples include pH and Electro-Conductivity (EC, Fig. 2, step 2), both measured *ex-situ* in the laboratory, even for lacustrine samples. These measurements were performed in the Montpellier laboratory using a HANNA Instruments HI991301 after a two-points calibration for pH and a single calibration for EC (mS/cm). Salinity, in terms of Total Dissolved Solids (TDS), was extrapolated from the EC following Rusydi (2018), Eq. (1):

$$\text{TDS} = \alpha \times \text{EC}_\text{T} \times 10^3 \tag{1}$$

with TDS in mg/L extrapolated from the EC at ambient temperature ($\text{EC}_\text{T}$) in mS/cm corrected by a conversion factor $\alpha \in [0.5; 0; 8]$ depending on the sample type (Rusydi, 2018). In Tibetan Plateau, the selected values are 0.65 or 0.8 (values from Supplementary Materials in Kou et al., 2022). Due to the wide range of salinity values among samples, it is mainly expressed in $\log_{10}$ (Kou et al., 2022). For new samples, a $\alpha$ of 0.65 was applied in this study to convert EC in TDS (i.e., in salinity). The



$EC_T$ is temperature compensated with Eq. (2):

$$EC_T = EC_{25\,°C} \times \beta T \tag{2}$$

where $\beta = 1.9\%$ is the temperature correction coefficient, $T$ is the temperature in degrees Celsius, and $EC_{25,°C}$ is the electrical conductivity standardized to 25 °C, as conventionally defined. For the salinity of samples published in Wang et al. (2021) and
Kou et al. (2022), please refer to the method section of both publications. In the ACADB, salinity values are available for 113 soil and 67 lacustrine samples.

Since only a few weather stations are available in ACA, extrapolated values from GIS databases were preferred to infer the climate parameter controlling the brGDGT distribution. Using GIS R packages (`rgdal`, version 1.6-7 and `raster`, version 3.6-30; Bivand et al., 2015; Hijmans et al., 2015), climate parameters were extracted from `worldclim2.1` (Fick and
Hijmans, 2017) and the extrapolated Aridity Index (AI) from the `CGIAR` database, (version 2; Trabucco and Zomer, 2018) for each surface sample from the ACA. The parameters used include Mean Annual Air Temperature (MAAT), Mean Annual Precipitation (MAP), and seasonal variables such as the mean air temperature of Months Above Freezing (MAF), as well as Mean Precipitation and Temperature for the Cold and Warm Quarters (MPCOQ, MPWAQ, MTCOQ, and MTWAQ).The AI is calculated using the formula Eq. (3):

$$AI = 10000 \times \frac{MAP}{MA[ET_0]} \tag{3}$$

where $MA[ET_0]$ represents the Mean Annual Reference Evapotranspiration (Trabucco and Zomer, 2018). It is noticeable that AI increases in humid environments and decreases in arid to hyper-arid systems. The thresholds and colour scale for aridity classes (hyper-arid, arid, semi-arid, dry sub-humid, and humid) used in this study are detailed in Table S.3 and follow the classification defined by Nash (1999).

**2.3 GDGT analytical methods**

BrGDGTs were analysed (Fig. 2, step 2) following the laboratory protocol fully detailed in Dugerdil et al. (2021a) and Davtian et al. (2018). First, we ground approximately one $cm^3$ of the soil or lacustrine samples in order to weigh them after a 24-hour lyophilization process. Then, total lipid content (TLC) was extracted twice from the sample by microwave-assisted extraction (MAE) at a temperature of 70 °C using DCM:MeOH (3:1, v/v). Following Huguet et al. (2006), a known concentration of an
internal standard ($C_{46}$ GTGT) was added to each TLC to estimate the absolute concentration of each GDGT compound. The TLC was separated into two fractions by elution on $SiO_2$ a column with hexane:DCM (1:1, v/v) and DCM:MeOH (1:1, v/v). The polar fraction containing br- and isoGDGTs was then dried under $N_2$ before being re-dissolved in hexane:iso-propanol (98:2) solvent prior to injection. Analyses were performed using a high-performance liquid chromatography coupled to mass spectrometry equipped with atmospheric pressure chemical ionization (HPLC/APCI-MS, Agilent 1260 Infinity coupled to a
6120 quadrupole mass spectrometer). The entire analytical process was carried out in the geochemistry laboratory LGLTPE at *ENS de Lyon*. GDGTs were detected using Single Ion Monitoring (SIM). The protonated molecules were detected at *m/z* 1302, 1300, 1298, 1296, 1292, 1050, 1048, 1046, 1036, 1034, 1032, 1022, 1020, 1018, and 744 ($C_{46}$). Finally, we manually





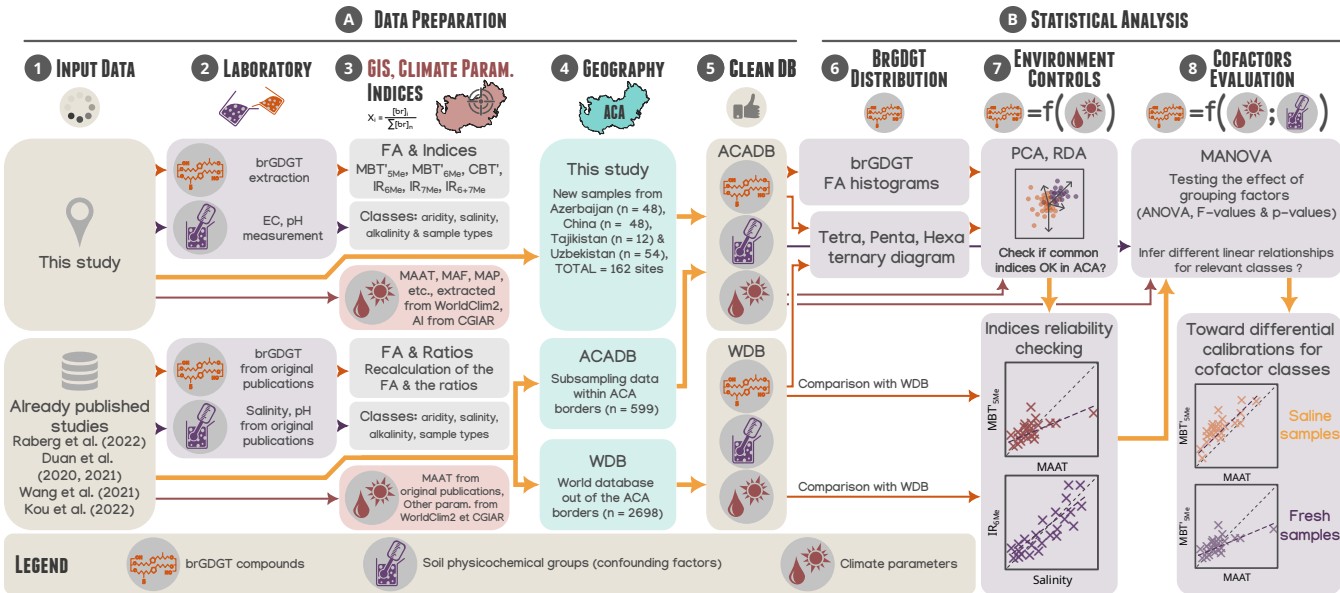

**Figure 2.** Methodological workflow followed in this study: from the first methodological step (input data – left-hand side) to the final results and perspectives. Each column of boxes represents a methodological step carried out for the data process (**A**) and the statistical analysis (**B**).

**Table 2.** Main GDGT indices discussed in this study with their formula, their main interpretations and their references.

| Index | Formula | Proxy interpretation | References |
|---|---|---|---|
| $\mathrm{MBT}'_{5Me}$ | $= \dfrac{Ia + Ib + Ic}{Ia + Ib + Ic + IIa + IIb + IIc + IIIa}$ | Temperature, soil moisture | De Jonge et al. (2014); Lin et al. (2024) |
| $\mathrm{MBT}'_{6Me}$ | $= \dfrac{Ia + Ib + Ic}{Ia + Ib + Ic + IIa' + IIb' + IIc' + IIIa'}$ | Temperature, independent from pH, better in drylands | De Jonge et al. (2014); Yang et al. (2015); Dang et al. (2016b) |
| $\mathrm{CBT}'_{5Me}$ | $= -\log\left(\dfrac{Ib + IIb}{Ia + IIa}\right)$ | pH, precipitation | De Jonge et al. (2014); Duan et al. (2022) |
| $\mathrm{CBT}'$ | $= -\log\left(\dfrac{Ic + IIa' + IIb' + IIc' + IIIa' + IIIb' + IIIc'}{Ia + IIa + IIIa}\right)$ | pH | De Jonge et al. (2014); Raberg et al. (2022b) |
| $\mathrm{IR}_{6Me}$ | $= \dfrac{\sum X_{6Me}}{\sum X_{5Me} + \sum X_{6Me}}$ | Salinity, pH | Raberg et al. (2022b) |
| $\mathrm{IR}_{7Me}$ | $= \dfrac{IIIa'' + IIa''}{IIIa + IIIa' + IIIa'' + IIa + IIa' + IIa''}$ | Salinity | Wang et al. (2021) |
| $\mathrm{IR}_{6+7Me}$ | $= \dfrac{IR_{6Me} + IR_{7Me}}{2}$ | Salinity | Wang et al. (2021) |
| $\mathrm{IR}'_{6+7Me}$ | $= \dfrac{0.5 \times (IIa' + IIb' + IIc' + IIIa' + IIIb' + IIIc') + IIIa''' + IIa'''}{IIa + IIb + IIc + IIIa + IIIb + IIIc + IIa' + IIb' + IIc' + IIIa' + IIIb' + IIIc' + IIIa''' + IIa'''}$ | Salinity | Wang et al. (2021) |



integrated each br- and isoGDGT based on the *m/z* ratio and retention time in order to identify each compound of brGDGTs with their 5-, 6- (De Jonge et al., 2014), and 7-methyls isomers (Ding et al., 2016). Following De Jonge et al. (2014), the
Roman numerals represent different GDGT structures. The 6-methyl brGDGTs are marked with an apostrophe after the Roman numerals to differentiate them from their 5-methyl isomers, and two apostrophes represent the 7-methyls (Ding et al., 2016). The measurement accuracy of the GDGT analysis method was assessed through the inter-calibration exercises conducted in 2023 (De Jonge et al., 2024b).

## 2.4 brGDGT indices calculation

Based on raw GDGT integrations, we calculated absolute concentrations expressed in $\mathrm{ng.g}_{\mathrm{sed.}}^{-1}$ (Huguet et al., 2006) and fractional abundances (FA; De Jonge et al., 2014) using a R routine (Fig. 2, step 3). The classical indices of methylation for 5- and 6-methyls ($\mathrm{MBT}'_{5\mathrm{Me}}$ and $\mathrm{MBT}'_{6\mathrm{Me}}$) and cyclisation (CBT' and $\mathrm{CBT}'_{5\mathrm{Me}}$), as well as, the isomer ratios ($\mathrm{IR}_{6\mathrm{Me}}$, $\mathrm{IR}_{7\mathrm{Me}}$, $\mathrm{IR}_{6+7\mathrm{Me}}$ and $\mathrm{IR}'_{6+7\mathrm{Me}}$; Wang et al., 2021) are also calculated and summarized in Table 2. To avoid overloading this study with multiple indices, we do not assess the Degree of Cyclisation index (DC; Sinninghe Damsté et al., 2009), nor its updated
version incorporating 5- and 6-methyl isomers (Baxter et al., 2019). Although the DC index more accurately reflects changes in the number of internal cyclopentane rings than the CBT index (which track both isomers and cyclisations), we focus solely on CBT, as it is more commonly used in brGDGT studies from drylands (Guo et al., 2021; Chen et al., 2021; Duan et al., 2022).

## 2.5 Database compilations

Two databases are compared in this study (Figs. 1 and 2, steps 4 and 5). The Arid Central Asian Data Base (ACADB, n = 761)
gathers new samples as well as samples collected from previously published studies, listed in Table 1 and Table S.2. Among them, the majority of the sites were already cleaned and homogenized by Raberg et al. (2022b). We appended the northern Iranian samples (Duan et al., 2020, 2022) and the Sibero-Mongolian samples that have already been published in Dugerdil et al. (2021a) as the New Mongolian–Siberian Database. This dataset gathers 43 different sites from the Baikal basin to the northern part of the Gobi Desert (Fig. 1B and geographical details on Fig. 1 from Dugerdil et al., 2021a). From Cromartie
et al. (2025), we appended 22 samples from Armenia, which follow an altitudinal gradient from the Ararat plain to the high plateau surrounding Lake Sevan (Fig. 1B). The salinity and 7-methyl FAs from Chinese data were also added (Wang and Liu, 2021; Kou et al., 2022). Among the ACADB (n = 761), there are 560 soil and 201 lacustrine samples. In order to compare the ACADB results, we also compiled a global Worldwide brGDGT surface Data Base (WDB, n = 2709) based on Raberg et al. (2022b) and Kou et al. (2022).

## 2.6 Statistical treatment

### 2.6.1 Univariate and multivariate analysis

Using R (version 4.4.2; R Core Team, 2020), we performed univariate linear relationships and multivariate analyses to understand environmental controls on brGDGT distributions (Fig. 2, step 7). The reliability of univariate relationships was inferred





by Pearson's $r$, coefficient of determination ($R^2$), adjusted-$R^2$ ($R^2_{\text{adj}}$), $p$-values, and Root Mean Square Error (RMSE). The

multivariate analyses were conducted with the `vegan` package on scaled data (version 2.6-8; Dixon, 2003) and included

Principal Component Analysis (PCA) on the brGDGT matrix and Redundancy Analysis (RDA) combining the brGDGT and

surface climate parameters matrixes (Dixon, 2003). PCA is an unconstrained ordination that reduces data dimensionality by

identifying axes (principal components) capturing the most variance. RDA is a constrained ordination that explains variation in

GDGTs using environmental variables. To meet the assumptions of linearity and normality required for both analyses, environ-

mental variables were standardized using the `scale()` function, while the FAs of each brGDGT compound were transformed

using the Hellinger transformation (Eq. 4), which down-weights dominant compounds.

$$\text{FA} - \text{transformed}_{\text{i,j}} = \sqrt{\frac{\text{FA}_{\text{i,j}}}{\sum\limits_{\text{j}} \text{FA}_{\text{i,j}}}} \tag{4}$$

with $\text{FA}_{\text{i,j}}$, the FA of compound $j$ in the sample $i$. Since the 7-methyl FAs are not available for all samples in the compiled

studies, they were removed from the databases for multivariate analysis. To select the most reliable environmental driving

factors to apply into the RDA, a Variance Factors analysis (VIFs, a method highlighting the covariance between factors) was

performed on climate parameters and soil characteristics (pH and TDS) using the `vif.cca()` function from the same R

package. To limit the covariance between them, only environmental factors below a threshold (e.g., below 10; Cao et al., 2014)

were kept for the RDA analysis and the following steps of the statistical workflow. The configurations of the two PCAs (for

soil and lacustrine samples) were compared using a Procrustes rotation analysis (i.e., comparing the similarity between PCA

and RDA ordination patterns by rotating one configuration to best match the other) and a PROTEST significance test (i.e.,

quantification of the fitting degree via permutation test) between the two PCAs using the package `vegan` (Dixon, 2003). The

same method was applied to compare the RDA brGDGT vs. climate parameters for soil and lacustrine samples. Finally, linear

relationships inferred between brGDGT indices and environmental factors follow Pearson's correlation (only coefficients of

determination with $p$-value < 0.001 are displayed on figures).

### 2.6.2 Grouping factor analysis


Samples were grouped by pH, aridity, salinity, and sample type to evaluate the most influential confounding factors. The applied

thresholds to bin classes are displayed in Table S.3. To identify data grouping patterns in relation to bioclimatic parameters,

sensitivity analyses were conducted in R by calculating the determination coefficients (multiple $R^2$ and $R^2$ for groups above

and below threshold values) across a continuous range of thresholds. For example, pH thresholds were tested from 4 to 11 in

0.01 increments. Multivariate Analysis of Variance (MANOVA) was performed with the `manova()` function to detect the most

important environmental factors driving the variance among the 15 brGDGT FA (only the 5- and 6-methyls were selected) and

among the main GDGT indices (Fig. 2, step 8). Then, the univariate ANOVA results were obtained with `summary.aov()`

for each brGDGT compound and index. Both functions come from the `stats` R base package (version 4.4.2; R Core Team,

2020). The MANOVA tests for differences in multiple dependent variables across different groups to see if group means are

significantly different, while the ANOVA tests for differences in the means of a few groups to determine if at least one group




mean is significantly different. The assumption of multivariate normality was tested with Mardia's Skewness and Kurtosis tests (i.e., MANOVA is possible only if the two *p*-values are higher than 0.05; Mardia, 1970). The assumption of homogeneity of variance-covariance was tested for each variable and each grouping factor using the Levene's test (Bierens, 1983). Using the most relevant confounding factors, specific $\mathrm{MBT}'_{5\mathrm{Me}}$-based temperature calibrations were done for each grouping factor. To

compare the linear relationship among groups, the significance of the difference was carried out with the *z*-statistic following Clogg et al. (1995), Eq. (5):

$$z = \frac{\beta_1 - \beta_2}{\sqrt{\mathrm{SE}_1^2 + \mathrm{SE}_2^2}} \tag{5}$$

with $\beta_n$ the coefficients and $\mathrm{SE}_n$ the standard errors of the linear regressions among the *n* groups. The *p*-values for the *z*-statistics are inferred with a normal distribution. The same *z*-statistic approach [Eq. (5)] was applied to determine the signifi-

cance of the difference between each linear model intercept (i.e., here, the offset between each calibration). The *z*-statistic was preferred to the t-test since the size of the data is high (e.g., more than 30 samples; Moore et al., 2009). All statistical treatments and graphical representations (except the map done with `QGIS 3.34 Pritzen` and the methodological workflow done with `Inkscape`) were performed in `R`. The plots were designed with the `ggplot2` package (version 3.5.1; Wickham, 2016) and, more particularly, the `ggtern` (version 3.5.0; Hamilton and Ferry, 2018) for the ternary diagram (Fig. 2, step 6).

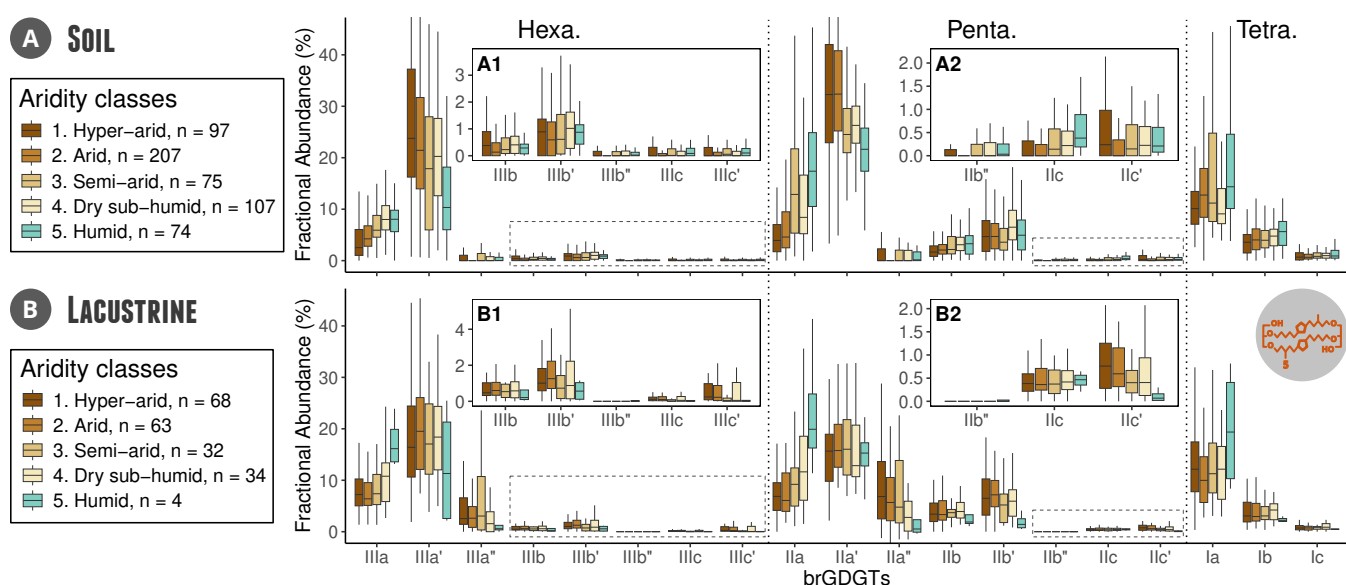

**Figure 3.** Distribution of individual brGDGTs grouped by aridity index, following the five aridity classes (Nash, 1999) in the ACADB for soil (**A**) and lacustrine samples (**B**). The brGDGT fractional abundances (FA) are displayed with ' for the 6-methyl and " for the 7-methyl isomers. The compounds of lowest abundances are zoomed on panels **A1**, **A2**, **B1** and **B2**.



# 3 Results

## 3.1 brGDGT distribution

### 3.1.1 brGDGT concentrations

The brGDGT concentration is heterogenous among the ACA surface samples, and it mainly depends on the sampling site location and sample type. They vary between $674 \pm 2825$, $87 \pm 314$ and $342 \pm 1822$ ng.g$_{\text{sed.}}^{-1}$ for lacustrine, soil and the whole database, respectively. Lacustrine samples are much richer in brGDGT than soil samples (about ten times more concentrated). Among the soil samples, the more moisture in the soils, the higher the concentrations, from $149 \pm 594$ to $24 \pm 33$ ng.g$_{\text{sed.}}^{-1}$. From drylands, samples from sand dunes are poorer than loess, silt-rich, and *solonchak* samples. They are close to the detection level (i.e., integration peaks smaller than twice the noise level), thus increasing uncertainties for indices based on 6- and 7-methyls.

### 3.1.2 brGDGT fractional abundances

With regard to FAs in soil samples, the prevalent compounds are IIa' (mean value $\simeq 30\%$), IIIa' ($\simeq 22\%$), Ia ($\simeq 14\%$), IIa ($\simeq 10\%$) and IIIa ($\simeq 6\%$, Fig. 3A). In lacustrine samples, the distribution is dominated by IIIa' ($\simeq 19\%$), IIa' ($\simeq 16\%$), Ia ($\simeq 12\%$), IIa ($\simeq 10\%$) and IIIa ($\simeq 9\%$, Fig. 3B). In contrast, compounds such as IIIb, IIIb', IIb'', IIIc, IIIc', IIb'', IIc, and IIc' are rare in both soil and lacustrine samples, with average abundances ranging from 1% to 2%. It is noteworthy that, for each compound, the 6-methyl isomers are more abundant than the 5-, and the 5- are more abundant than the 7-. This trend is more marked for soil than lacustrine samples. The 7-methyl isomers have higher importance in lacustrine than in soil samples. For both sample types, the histograms reveal that brGDGT distributions vary across the different aridity classes, with trends emerging with increased aridity: IIIa increases in the humid group, while IIIa' and IIIa'' are higher in the arid and hyper-arid groups. A similar trend is observed between IIc and IIc'. Additionally, IIa increases with humidity, while the IIa' distribution remains largely insensitive to changes in aridity classes for lacustrine but decrease with higher humidity for soil samples. Compounds Ia, Ib and Ic exhibit discernible variations between aridity classes, although the observed trends are not unequivocal. Finally, aridity control is less evident in other low-abundance compounds, including the IIIb, IIIc, and IIb, and all 7-methyl isomers.

### 3.1.3 Methylation distribution

In soil samples from the ACADB, tetra-, penta-, and hexamethylated brGDGTs range from 0 to 55%, 20 to 80%, and 0 to 85%, respectively (Fig. 4A1 and A2). The distribution of hexamethylated compounds is the most variable in ACA. By contrast, in the WDB, brGDGT distributions are strongly centred around tetramethylated compounds, with only a few samples showing high hexamethylated fractions; the majority contain less than 20% of hexamethylated compounds. About the aridity effect, ACADB samples from a humid environment fit the WBD distribution better, while arid and hyper-arid samples shift towards lower tetramethylated content.





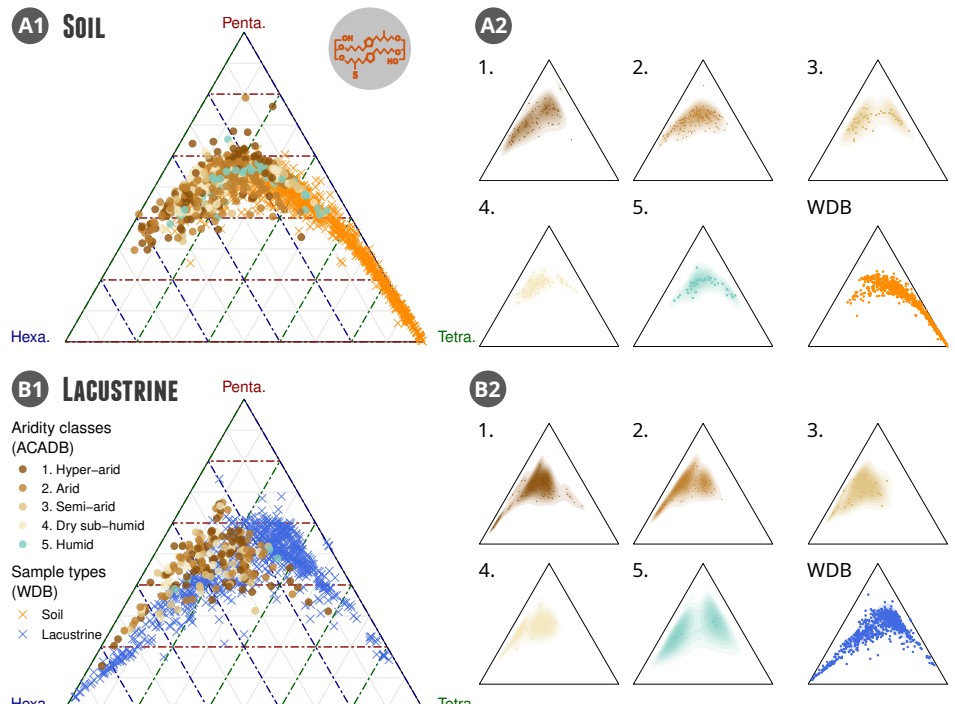

**Figure 4.** Ternary diagram showing the relative importance of tetra-, penta-, and hexamethylated brGDGT compounds for (**A**) soil samples and (**B**) lacustrine samples. ACADB samples (dots) are compared to WDB ones (crosses). The plots are displayed by aridity classes. Panels **A1** and **B1** overlay the ACADB and WDB, while panels **A2** and **B2** distinctly present the distribution of each aridity class for soils (**A**) and lacustrine samples (**B**).

Similarly, for lacustrine samples (Fig. 4B1 and B2), the ACADB shows tetramethylated brGDGTs ranging from 0 to 60%, pentamethylated from 10 to 85%, and hexamethylated from 10 to 90%. Compared to WDB, the *croissant shape* of the ternary distribution is retained but shifts towards less tetra- and more pentamethylated forms. For lacustrine samples, the aridity effect seems mitigated compared to soil samples. However, the samples from humid environments are spread along a bimodal distribution following the hexamethylated axis, while this is not the case for hyper-arid to dry sub-humid samples with hexam-
ethylated above 20%.

### 3.1.4   brGDGT multivariate spaces

PCAs were performed on 400 soil and 361 lacustrine surface samples, using Hellinger-transformed FA values (Fig. 5A and B). The brGDGT loadings for the first and second components explain 42% and 18% for soil samples and 40% and 19% for lacustrine samples, respectively. For both, the most important contributing compounds are IIIa', IIa, IIa', and Ia, followed by
IIb', IIIa, IIIb', IIc', and IIIc'. $PCA_1$ corresponds to a gradient between the 5- and 6-methyl isomers, with positive loadings for IIa, Ia, Ib, and IIIa and negative loadings for IIa', IIIa', and IIb'. $PCA_2$ follows a gradient marked by the number of internal





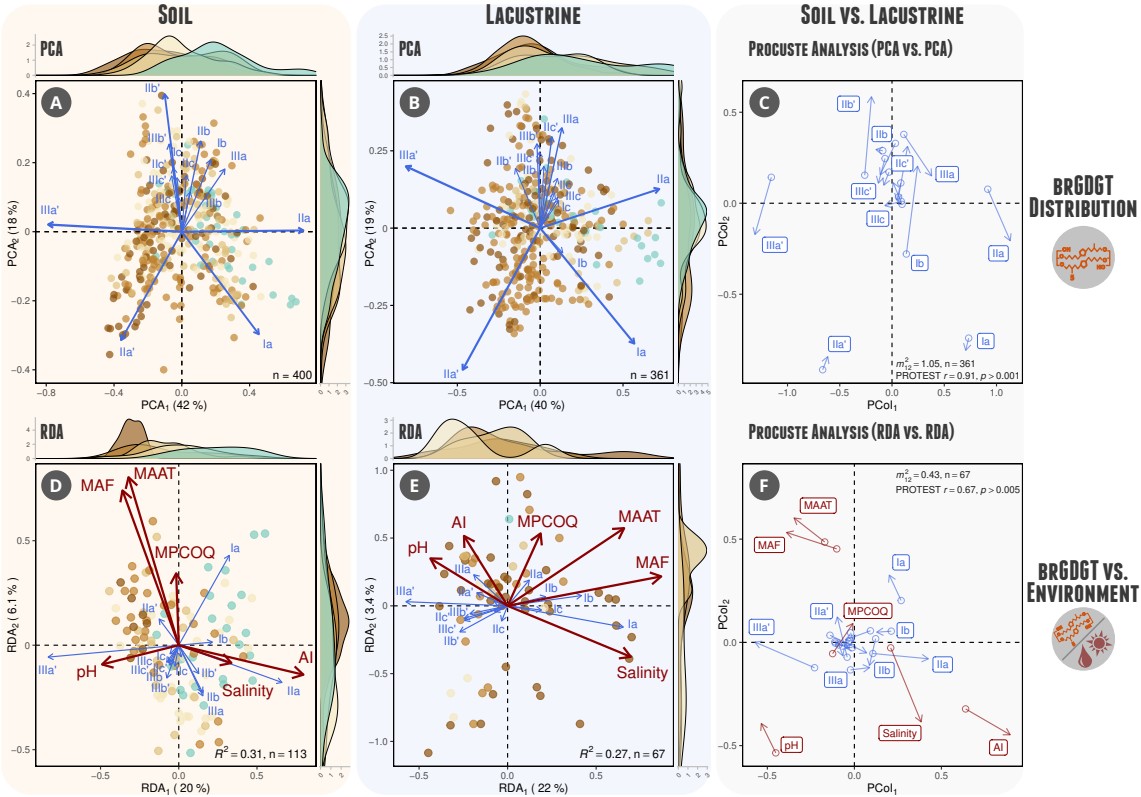

**Figure 5.** Multivariate analyses of the ACA brGDGT surface samples highlighting the brGDGT distribution using Principal Component Analysis (PCA) for soil (**A**) and lacustrine (**B**) samples, and (**C**) the Procrustes rotation analysis to compare the distribution of the brGDGTs along the loadings of the two PCAs (i.e., multivariate configurations similarity assessment). Then, Redundancy Analyses (RDA) track the main environmental drivers of soil (**D**) and lacustrine (**E**) sample distribution, and the Procrustes rotation analysis compares them (**F**). Analyses were performed only on the 5- and 6-methyl due to the few number of 7-methyl measurements in the database. The most contributing environmental drivers (Aridity Index, salinity, mean air temperature of Months Above Freezing, Mean Annual Air Temperature, Mean Precipitation of the Cold Quarter, pH and Altitude) were selected using a Variance Factor test (i.e., VIF < 10 for all). The PCA is performed on the whole ACA dataset ($n_{soil}$ = 400 and $n_{lac.}$ = 361), while the RDA only covers 113 soil and 67 lacustrine samples due to scarce pH and salinity measurements available. The colour code for dots corresponds to the five aridity classes.

cyclisation: the positive loading is driven by compounds with one or two internal cyclisations (mainly IIb', IIIb', IIIc', and IIc'), while the negative loading presents compounds without internal cyclisation (Ia and IIa'). About the distribution of aridity classes over this multivariate space (highlighted by the upper and lateral sample densities in Fig. 5), a clear aridity gradient can

be superimposed on the isomer gradient (i.e., along $PCA_1$ with humid samples on a positive loading and hyper-arid samples on a negative one). This gradient is clearer for soil than for lacustrine samples. In between, arid, semi-arid, and dry sub-humid samples show a higher internal cyclisation number (positive $PCA_2$), giving a *triangular* shape to the sample distribution.



Procrustes rotation analysis, along with the PROTEST *r* and *p*-values, was used to assess the similarity between the two PCAs (Fig. 5C). The statistical results reveal that the brGDGT multivariate space is very similar for soil and lacustrine samples

with $m_{12}^2 = 1.05$, $r = 0.91$ and *p*-value < 0.01. The more variable components between soil and lacustrine multivariate spaces are IIIa', IIa, and Ib. It is noteworthy that Ib is correlated with IIb and IIIa in soils, while it is correlated to Ia in lacustrine samples.

### 3.2   brGDGT responses to environmental controls

#### 3.2.1   ACA bioclimate multivariate space

When considering the bioclimate space of ACA sampled at the ACADB sampling points (Fig. 1E), the main loading ($PCA_1$ = 53%) is a temperature gradient and the secondary loading ($PCA_2$ = 29%) is a precipitation gradient. About the climate parameter contribution, AI and MAP have more influence on the ACA bioclimate variance, followed by MPCOQ and MAAT. In ACA, all temperature parameters are strongly correlated, indicating that temperature seasonality is roughly homogeneous within the ACADB. This observation is consistent with the lower variance explained by MAF than MAAT. The coefficient

of variation among the ACADB is higher for MAAT than seasonal temperature parameters (170 vs. 30%). Contrarily, Mean Precipitation of Warm / Cold Quarters (MPWAQ and MPCOQ, respectively) are negatively correlated, showing a split in seasonal precipitation patterns among the ACADB, which is in line with the global climate observations in ACA (Chen et al., 2024).

In order to verify that these parameters remain the primary factors to explain the brGDGT variance space, Variance Factor

(VIF) analyses were performed on two models: (1) using all environmental parameters (i.e., all climate parameters, altitude, pH, and salinity) and (2) only a selection of climate parameters to verify that VIFs < 10 (Cao et al., 2014). In Table S.4, the multicollinearity between MAAT, MTCOQ, and MTWAQ is clear (VIFs > 159.1) and reduced with MAF (VIF = 26.7). Multicollinearity is lower among the precipitation parameters (VIF < 37.8). When keeping only AI, MAAT, MAF, MPCOQ, pH, and salinity, all VIFs are below 6. VIFs for pH and salinity do not change between the two models, showing their inde-

pendence from climate parameters. Finally, from the total set of environmental variables, the VIF analysis removed the altitude, MAP, MPCOQ, MTCOQ, and MTWAQ. The remaining environmental parameters have $VIF_{MAAT} = 6$, $VIF_{MAF} = 4.5$, $VIF_{MPCOQ} = 2.1$, $VIF_{AI} = 1.5$, $VIF_{pH} = 1.5$ and $VIF_{Salinity} = 1.3$.

### 3.2.2   Controls on brGDGTs in the ACA bioclimate space

Regarding the environmental controls on brGDGT distribution, RDAs were performed based on analyses of 113 soil and 67

lacustrine samples (excluding samples without pH and salinity data), including 15 brGDGT compounds (limited to 5- and 6-methyl isomers) and six environmental variables (Fig. 5D and E). The first two RDA axes explained 20% and 5.9% of the variance in soil samples, and 22% and 3.4% in lacustrine samples. The overall correlation between the brGDGT composition and the environmental variables was 0.31 for soils and 0.27 for lakes. About the distribution of sites and brGDGT vectors,





we observe the conservation of the distribution between PCA and RDA for soil (Fig. 5A and D): similar $PCA_1$ and $RDA_1$
loadings, and reverse loading among $RDA_2$ and $PCA_2$.

The distribution of soil samples in the RDA ordination reflects a gradient of aridity, with humid samples on positive loading associated with high AI and salinity and hyper-arid samples on negative one correlated with pH, MAF, and MAAT. This gradient corresponds to a predominance of GDGT compounds IIIa', IIIb', and IIa' in arid samples, while more humid sites are associated with higher FAs of IIa and Ia. Thus, the aridity gradient along the $RDA_1$ loading is clearly conserved from PCA
to RDA. Surprisingly, higher salinity is associated with higher AI (i.e., humid conditions) and negatively correlated with pH (i.e., acidic conditions). Although soil salinity is expected to increase under bioclimatic aridity (Muhammad et al., 2008), our results show that in ACA, salinity is largely independent of climate parameters (Fig. 1E).

For lacustrine samples, the loadings are quite different between PCA and RDA, mainly due to low IIa and IIa' correlations with environmental parameters in RDA (despite their strong importance in primary and secondary loadings in PCA). The
aridity gradient is not clear (mainly due to the scarcity of hyper-arid and humid lacustrine samples). Here, salinity is negatively correlated with pH and AI. Additionally, tetramethylated compounds (Ia, Ib, and Ic) are controlled by both temperature (MAAT and MAF) and salinity.

The Procrustes rotation analysis carried out on the two different RDAs (Fig. 5F) shows that the correspondence between the two RDA spaces is smaller than between the two PCA spaces (Fig. 5C) with $r = 0.67$ and $p$-value $< 0.003$. Compounds IIIa
and IIa exhibit the most pronounced rotations (i.e., the differences in their loadings between the two PCAs as revealed by the Procrustes rotation analysis), indicating substantial differences in environmental controls between soil and lacustrine samples. For environmental parameters, salinity and AI are the two parameters with the highest degree of rotation between the soil and lacustrine RDA spaces. AI is more contributing in the soil samples RDA, while salinity is more contributing in the lacustrine samples RDA.

### 3.2.3 Methylation indices relationships to climate parameters

Linear relationships between the two main indices of methylation generally used in brGDGT calibration studies, the $MBT'_{5Me}$ and $MBT'_{6Me}$, are tested against pH, AI, MAAT, and salinity (Fig. 6). The $R^2$ are given for each relationship and for both subsets of sample type (i.e., soil and lacustrine samples). For soil samples, the strongest $MBT'_{5Me}$-based linear relationship is related to MAAT ($R^2 = 0.31***$), even if it is also associated with pH ($R^2 = 0.17***$) and AI ($R^2 = 0.15***$). The relationship
with salinity is, however, not significant. Contrarily, for lacustrine samples, salinity exhibits the strongest relationships with $MBT'_{5Me}$ ($R^2 = 0.31***$). Correlation coefficients obtained with $MBT'_{6Me}$ are generally lower than those with $MBT'_{5Me}$ with pH for soils ($R^2 = 0.11***$) and with MAAT for lacustrine samples ($R^2 = 0.13***$). When statistically significant, all relationships follow similar trends (positive or negative) for $MBT'_{5Me}$ and $MBT'_{6Me}$.

Comparing the ACADB to the WDB, some relationships are similar: $MBT'_{5Me}$ with MAAT and salinity, and $MBT'_{6Me}$ with
pH and AI, despite the tighter climatic range of ACADB compared to WDB. Contrarily, some trends are reversed: $MBT'_{6Me}$ with salinity and $MBT'_{5Me}$ with pH. It is also noticeable that ACADB samples from humid and dry sub-humid conditions fit better with the WBD distribution than arid and semi-arid systems. Hyper-arid samples often have extreme values.




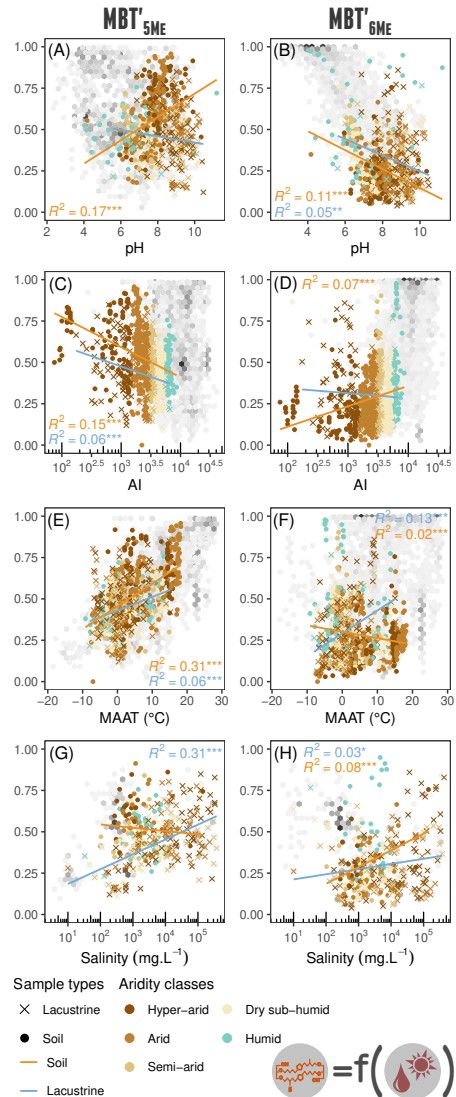

**Figure 6.** $\mathrm{MBT}'_{5\mathrm{Me}}$ and $\mathrm{MBT}'_{6\mathrm{Me}}$ relationships with pH (**A** and **B**), Aridity Index (**C** and **D**), MAAT (**E** and **F**), and salinity (**G** and **H**). The colour refers to the aridity classes for the samples. Two groups per sample type (i.e., lacustrine and soil) are used to infer linear relationships. The grey hexagonal bins show the sample density from the WDB. $R^2$ values are shown only for statistically significant regressions ($p<0.05$), while $p$-values below 0.001, 0.01, and 0.05 are indicated with \*\*\*, \*\*, and \*, respectively.

### 3.2.4 Cyclisation, isomerisation, and pH

The cyclisation and isomer indices (here, CBT', $\mathrm{CBT}'_{5\mathrm{Me}}$, $\mathrm{IR}_{6\mathrm{Me}}$ and $\mathrm{IR}'_{6+7\mathrm{Me}}$), which are commonly considered to be pH-
related proxies, are tested against aridity, pH and salinity on WBD and ACADB. On the ACADB, the linear relationships between pH and these indices are not significant for lacustrine samples (Fig. 7). In soil samples, isomer ratios appear to





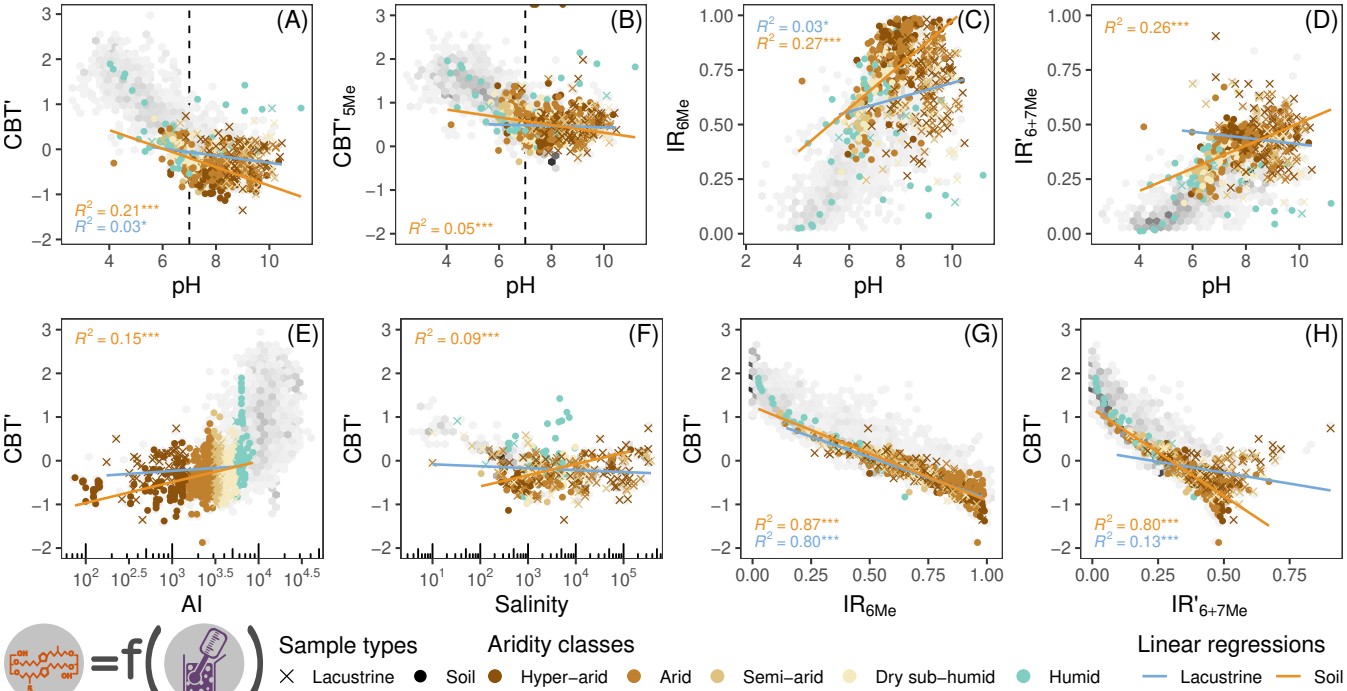

**Figure 7.** pH-related indices, mainly cyclisation indices (CBT' and $\mathrm{CBT}'_{5Me}$) and isomer ratios ($\mathrm{IR}_{6Me}$ and $\mathrm{IR}'_{6+7Me}$) are shown. Linear regressions are tested with pH (**A** to **D**), AI (**E**) and salinity (**F**) and among them (**G** and **D**). The colours refer to the aridity classes for the samples. Two groups by sample type (i.e., lacustrine and soil) are used to infer linear relationships. The grey hexagonal bins show the sample density from the WDB. AI and salinity are displayed on $\log_{10}$ scale. $R^2$ values are shown only for statistically significant regressions ($p<0.05$), while $p$-values below 0.001, 0.01, and 0.05 are indicated with ***, **, and *, respectively.

correlate more strongly with pH than cyclisation indices, with $\mathrm{IR}_{6Me}$ showing the highest explanatory power ($R^2 = 0.27$***). This pattern is consistent in both the WDB and ACADB datasets. About the cyclisation, CBT' is more linearly related to pH than $\mathrm{CBT}'_{5Me}$. However, both indices suffer from a relation break around a pH threshold of 7.3 (highlighted by dashed lines on Fig. 7A and B). This indicates that the cyclisation degree is linearly correlated to pH only in acidic samples. To test the pH threshold value, the CBT' vs. pH linear relationship was tested for the two groups, below and above the threshold, using a continuous implementation of threshold pH from 4 to 11, with 0.01 steps each (Fig. S.2). The best $R^2$ is for a pH of 7.3, with multiple $R^2$ of 0.45 for overall soils, 0.54 for acidic soils, and 0.05 for alkaline soils. Similar thresholds are also observable with aridity and salinity, although with lower regression strength ($R^2$ of 0.15*** and 0.09***, respectively). Finally, there is a strong similarity between CBT'-$\mathrm{IR}_{6Me}$ and CBT'-$\mathrm{IR}'_{6+7Me}$ ($R^2$ of 0.87*** and 0.80***) for soils, while this does not hold true for lacustrine samples (Fig. 7G and H).





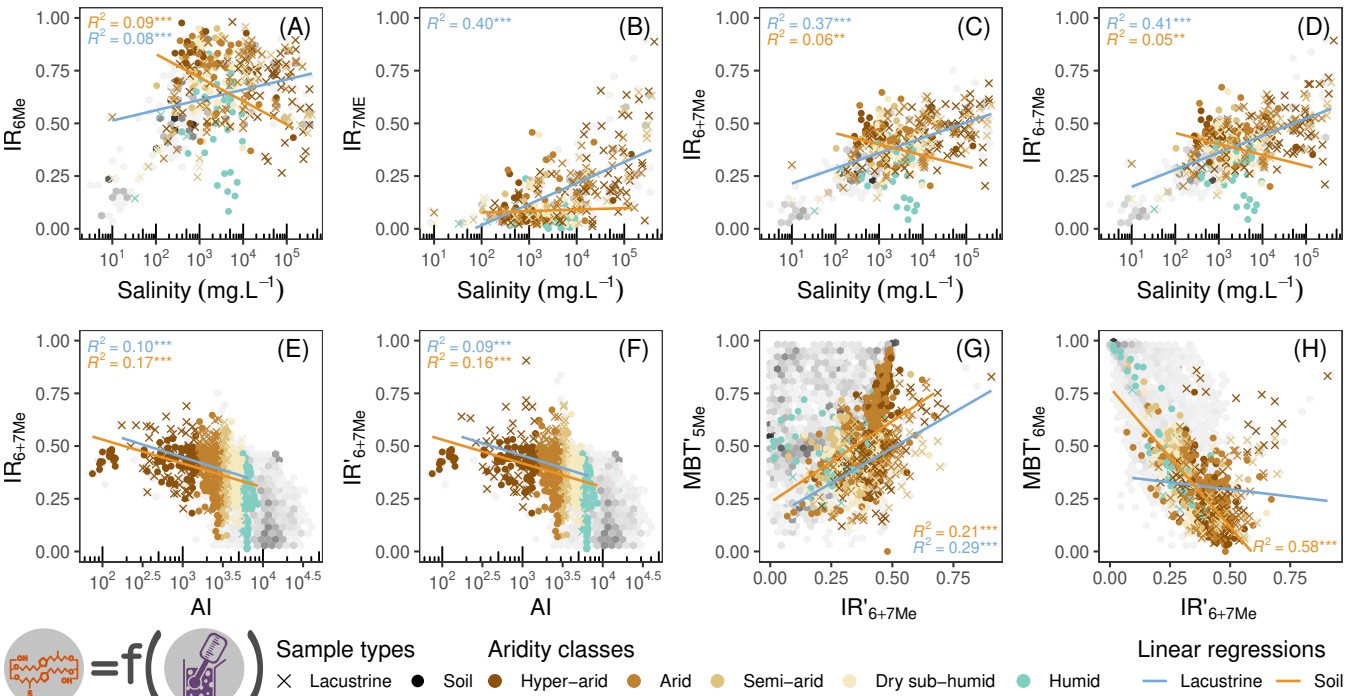

**Figure 8.** Salinity-related indices: $IR_{6Me}$, $IR_{7Me}$, $IR_{6+7Me}$ and $IR'_{6+7Me}$ are the most ubiquitous brGDGT-based indexes used to infer salinity in ACA. The indices are compared to salinity (**A** to **D**), the Aridity Index (**E** and **F**), $MBT'_{5Me}$ and $MBT'_{6Me}$(**G** and **H**). The colours refer to sample aridity classes. Two groups per sample type (i.e., lacustrine and soil) are used to infer linear relationships. The grey hexagonal bins show the sample density from WDB. AI and salinity are displayed on a $\log_{10}$ scale. $R^2$ values are shown only for statistically significant regressions ($p<0.05$), while $p$-values below 0.001, 0.01, and 0.05 are indicated with ***, **, and *, respectively.

### 3.2.5 Isomer ratios responses to aridity and salinity

The common isomer ratios used to infer salinity are tested with aridity and salinity (Fig. 8). The linear relationships between IRs and $MBT'_{6Me}$ are also tested. For salinity, the correlations are more statistically significant with lacustrine than with soil samples, by order of strength: $IR'_{6+7Me}$ ($R^2 = 0.41$***), $IR_{7Me}$ (0.40***) and $IR_{6+7Me}$ (0.37***). For soil, $IR_{6Me}$ has the strongest relationship with pH ($R^2 = 0.09$***), but the relationships with aridity are also significant, especially with $IR_{6+7Me}$ ($R^2 = 0.17$***). The relationships are comparable between ACADB and WDB, although the WDB environmental range is wider, especially for low salinity and high AI values. $IR'_{6+7Me}$ is slightly correlated with the $MBT'_{5Me}$ index in lacustrine samples ($R^2 = 0.29$***), while it is more significantly correlated to $MBT'_{6Me}$ in soil samples (0.58***). These relationships are similar in WDB, although no linear relationship appears between $IR'_{6+7Me}$ and $MBT'_{5Me}$ in WDB. Moreover, the isomer ratio relationship with salinity drastically changes depending on the relative weight given to 7-methyl isomers. The regression between IRs and salinity shows increasing explanatory power with 7-methyls weight, with $R^2$ values rising from 0.08 for $IR_{6Me}$





to 0.40 for $\mathrm{IR_{7Me}}$, and from 0.37 for $\mathrm{IR_{6+7Me}}$ to 0.41 for $\mathrm{IR'_{6+7Me}}$. This is mainly due to the salinity's positive correlation with 7-methyls and negative one with 6-methyls (respectively, Pearson's $r$ of 0.64 and -0.35, Fig. S.3).

### 3.3 Robustness of the analysis of variance between groups

Since it was previously shown that the different environmental parameters have different interactions in the brGDGT distribution worldwide and in the ACA (Yang et al., 2014; Deng et al., 2016), both on soil and lacustrine samples, we evaluate the strength of each driving factor. Since the *sample type* group is a qualitative factor, the other quantitative factors (i.e., bioclimate and physicochemical parameters) were also binned into qualitative groups (see Table S.3 for the thresholds used for binning). The two MANOVA models inferred to evaluate their differential influences on (1) FAs distribution and (2) indices from the ACADB are displayed in Table 3.

#### 3.3.1 Analyses of variance for the FAs

This first MANOVA tests the response of the 15 brGDGT FAs to the four grouping factors: aridity, pH, salinity, and sample type. MANOVA results (in terms of $F$-statistics and $p$-values) are given in the first row of Table 3 while ANOVAs for each compound is given in rows two to 16. Levene's test $p$-values are provided in Table S.5. The higher MANOVA $F$-statistic (with a $p$-value < 0.001, symbolized by the *** symbol) represents the stronger environmental parameter to separate FA into groups. First, it appears that the sample type ($F = 10.2$***) is the most influential grouping factor on the FA variance, followed by salinity (7***) and pH (6.5***). Moreover, the sample type is responsible for the clustering among IIIa, IIIb, IIb, and Ic, while pH mainly influences IIa', IIa, Ic, and IIIa', and salinity influences IIIa, IIIa', IIIc, and IIb'. Aridity plays a role mainly on IIa, Ia, and IIIa'. Generally, the main compounds are impacted in their distribution by all confounding factors.

#### 3.3.2 Analyses of variance for the brGDGT-based indices

Since the majority of brGDGT applications are based on traditional indices, the response of $\mathrm{MBT'_{5Me}}$, $\mathrm{MBT'_{6Me}}$, $\mathrm{IR_{6Me}}$, $\mathrm{IR_{6+7Me}}$ and $\mathrm{IR'_{6+7Me}}$ to aridity, pH, salinity, and sample type was tested in the second MANOVA (line 16 of Table 3 followed by the ANOVA statistical results for each index). For MANOVA, the most important grouping factor is salinity ($F = 9.1$***), followed by pH and sample type (8.4***). Aridity is the weakest grouping factor (5.7***). The graphical representation of the variance for each group is available in Fig. S.4. More specifically, for $\mathrm{MBT'_{5Me}}$, the variance is mainly explained by sample type (37.1***) and salinity (17.7***). For $\mathrm{MBT'_{6Me}}$, the variance is mainly explained by salinity (13***) and aridity (11.9***). IR indices show stronger clustering based on aridity and salinity than on pH or sample type. Aridity and salinity gradients are clear among groups for $\mathrm{IR_{6+7Me}}$ and $\mathrm{IR'_{6+7Me}}$ while $\mathrm{IR_{6Me}}$ variance is more steady. For all three indices, there is a clear distinction for hypersaline and humid group variance (low index value for humid groups and high value for hypersaline). $\mathrm{CBT'_{5Me}}$ have a similar variance for each group, while CBT' is clustered by aridity and pH (13.7*** and 13.4***).





**Table 3.** Statistical results (approximate *F*-statistics and *p*-values) of the two MANOVAs carried out to test the brGDGT (1) FAs and (2) indices responses to environmental classes (i.e., aridity, pH, salinity, and sample type). The MANOVA *F*-statistics are presented in the first row of the two models, followed by the *F*-statistics of the univariate ANOVA for each item (i.e., each FA and each index). The significance is given by the number of stars[a]. The samples without pH or salinity measurements were removed from the analysis (i.e., n = 328). The *p*-values from Levene's test, which are important for assessing the reliability of the MANOVAs, are provided in Table S.5.

| Model | Compound | Aridity | pH | Salinity | Sample type |
|---|---|---|---|---|---|
| **Model 1** | **F-statistics** | 3.4*** | 6.5*** | 7*** | **10.2***[b]** |
| **(FAs)** | f(IIIa) | 1.7 | 3.6* | **29***** | **38.2***** |
| | f(IIIa') | 10.8*** | 9.6*** | 14.7*** | 6.5* |
| | f(IIIb) | 0.7 | 2.4 | 1.4 | 25.8*** |
| | f(IIIb') | 2.3 | 2.8 | 1.2 | 1.4 |
| | f(IIIc) | 1.2 | 6.1** | 9.8*** | 1.7 |
| | f(IIIc') | 0.9 | 1.9 | 2.7* | 0.4 |
| | f(IIa) | **22***** | 15.7*** | 2.9* | 1.5 |
| | f(IIa') | 2.2 | **18.5***** | 13.9*** | 4.6* |
| | f(IIb) | 0.6 | 4.4* | 2.5 | 20.2*** |
| | f(IIb') | 1.3 | 11.5*** | 6.6*** | 10.3** |
| | f(IIc) | 1.7 | 1.8 | 0.8 | 0.5 |
| | f(IIc') | 1 | 5** | 1.5 | 1 |
| | f(Ia) | 18.5*** | 6.7** | 4.7** | 4.1* |
| | f(Ib) | 1.4 | 2.2 | 2.5 | 0.9 |
| | f(Ic) | 3.7** | 12.7*** | 0.8 | 13.9*** |
| **Model 2** | **F-statistics** | 5.7*** | 8.4*** | **9.1***** | 8.4*** |
| **(Indices)** | $MBT'_{5Me}$ | 6*** | 7.2*** | 17.7*** | **37.1***** |
| | $MBT'_{6Me}$ | 11.9*** | 8*** | 13*** | 1.3 |
| | $IR_{6Me}$ | 13*** | 11.3*** | 9.9*** | 5.4* |
| | $IR_{6+7Me}$ | 24.8*** | 6.1** | 13.9*** | 1 |
| | $IR'_{6+7Me}$ | **28.1***** | 3.5* | **18.9***** | 0.5 |
| | CBT' | 13.7*** | **13.4***** | 8.7*** | 2.3 |
| | $CBT'_{5Me}$ | 5*** | 4.8** | 0.6 | 4.4* |

[a] The *p*-values are expressed in terms of *stars* with *** for $p \leq 0.001$, ** $p < 0.01$, * $p < 0.05$ and nothing for not-significant *F*-statistic (i.e., *p*-value $\geq 0.05$). [b] The highest *F*-statistic for each MANOVA and ANOVA is displayed in bold text.

## 4 Discussion

Comparing the ACADB results with other studies, the question of the relative importance of the confounding factor on past applications from drylands is raised. The effects of the confounding factors, such as pH (mainly for extrema; Duan et al., 2020),
aridity, seasonality, and salinity, are currently studied in ACA (Guo et al., 2021; Chen et al., 2021; Kou et al., 2022; Duan et al., 2022). More generally, the impact of confounding factors on geochemical proxies used to reconstruct palaeoenvironmental changes has been increasingly recognized in brGDGT-based temperature calibrations (De Jonge et al., 2014; Häggi et al., 2023). Mainly, based on the ACADB results, we discuss (1) the impact of these factors on brGDGT indices (methylation, isomer and cyclisation indices); then (2), the complex interaction between confounding factors; (3) the applicability of former
and new calibration depending on confounding factor classes; and (4), we provide recommendations for their applicability in



the past brGDGT archives in drylands. We will first focus on the reliability of $\mathrm{MBT'_{5Me}}$ and $\mathrm{MBT'_{6Me}}$ to reconstruct past MAAT and on isomer or cyclisation indices to infer salinity and pH in drylands. The following discussion is mainly based on ACADB results from linear regressions, as well as multivariate and variance analyses. However, due to limited metadata availability, these integrative statistical approaches (RDAs and MANOVAs) were applied to a reduced dataset (113 soil and 67 lake samples), and further research is needed to confirm the conclusions drawn.

## 4.1 Applicability of brGDGT-based proxies

### 4.1.1 $MBT'_{5Me}$ responses to temperature

The analysis of the ACADB climate space, based on the PCA (Figure 1E) and RDAs (Figure 5D and E), indicates that MAAT better captures both the climate variability and the brGDGT response across ACA compared to MAF. In both datasets, the proportion of variance explained by MAAT exceeds that explained by MAF, with the only exception being brGDGT assemblages from lacustrine samples. Based on this, we focus the discussion on MAAT, even though MAF is often preferred in brGDGT studies due to its relevance for representing the bacterial growing season (Deng et al., 2016; Dearing Crampton-Flood et al., 2020). However, the actual timing of bacterial growth may depend not only on temperature, but also on soil water availability (Lei et al., 2016). This is particularly relevant in the ACADB region, where MAF generally aligns with summer across ACA, while soil moisture availability is not spatially synchronous, with rainfall peaks in spring and autumn in eastern ACA, and in summer in the western part (Figure S.1).

When considering the $\mathrm{MBT'_{5Me}}$-MAAT relationships in the ACADB, the determination coefficients for both soil and lacustrine samples are limited (Fig. 6). The linear correlation is significantly higher in semi-arid to humid soils than hyper-arid and arid soils (Figs. S.7), in line with Wang et al. (2019). This attenuation of the $\mathrm{MBT'_{5Me}}$-MAAT correlation is also well observed on lacustrine samples where $\mathrm{MBT'_{5Me}}$ is more correlated to salinity than to MAAT (Fig. 6E and G) in line with Liang et al. (2024). In drylands, $\mathrm{MBT'_{5Me}}$ is also correlated with soil water content (Dang et al., 2016b). However, numerous temperature reconstructions use the $\mathrm{MBT'_{5Me}}$ index to capture the brGDGT compound's response to temperature (De Jonge et al., 2014; Chen et al., 2021), even if several studies have shown a strong bias in $\mathrm{MBT'_{5Me}}$-temperature relationship under arid conditions (Sun et al., 2019; Dugerdil et al., 2021a). Guo et al. (2021) demonstrated that in arid environments, the relationship between temperature and brGDGT methylation differs between 5- and 6-methyl isomers. The dominance of Actinobacteria and Verrucomicrobia, each linked to distinct 5- and 6-methyl brGDGT signatures, in arid soils may help explain the limited effectiveness of the $\mathrm{MBT'_{5Me}}$ index in capturing climate signals within the ACADB. In this context, $\mathrm{MBT'_{5Me}}$-based temperature calibration in drylands may have reduced reliability. Among solutions to improve this type of calibration, specific $\mathrm{MBT'_{5Me}}$-based temperature calibration can be provided for specific confounding factor classes.

### 4.1.2 $MBT'_{6Me}$ responses to climate

$\mathrm{MBT'_{6Me}}$-MAAT trends opposite for the WDB compared to the ACADB (Fig. 6F), even if the correlations with temperature remain weak. The correlation is slightly better with AI (Fig. 6D). The $\mathrm{MBT'_{6Me}}$ index has been proposed as a reliable temper-




ature proxy when MAP and MAAT are negatively correlated (Guo et al., 2021), a condition not met in the ACADB dataset. From the ACADB, we can estimate that $\mathrm{MBT}'_{6\mathrm{Me}}$ is slightly more controlled by AI than MAAT. Moreover, the tetramethylated

compounds (which are the major compounds involved in $\mathrm{MBT}'_{5\mathrm{Me}}$ and $\mathrm{MBT}'_{6\mathrm{Me}}$, De Jonge et al., 2014) are not as important in ACADB as in WDB (Fig. 4A). Additionally, the climate response of tetramethylated compounds is not clear in ACADB (Fig. 5D), while the IIa' is well correlated with MAAT. Initially designed by De Jonge et al. (2014) as a methylation index for tetramethylated over the sum of tetramethylated plus 6-methyl isomers, the $\mathrm{MBT}'_{6\mathrm{Me}}$ is understudied. However, Wang et al. (2016) and Guo et al. (2021) observed in arid soils that the $\mathrm{MBT}'_{6\mathrm{Me}}$ has a better response to temperature and aridity changes

than the $\mathrm{MBT}'_{5\mathrm{Me}}$. Also, their studies show an opposite correlation with MAAT than in global soil datasets, in lines with our results.

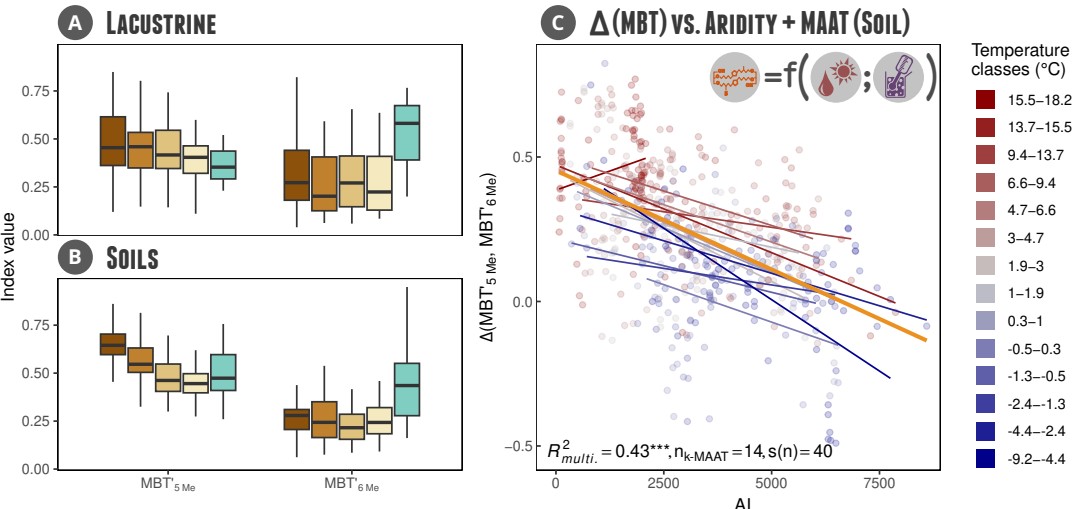

**Figure 9.** Illustration of the approach comparing $\mathrm{MBT}'5\mathrm{Me}$ and $\mathrm{MBT}'6\mathrm{Me}$ to investigate interactions between MAAT, aridity, and sample type. Boxplots of both indices are shown by aridity classes (see Fig.1 for colour codes), separately for soil (**A**) and lacustrine (**B**) samples. Panel **C** displays the relationship between $\Delta(\mathrm{MBT}'5\mathrm{Me}, \mathrm{MBT}'6\mathrm{Me})$ and the Aridity Index for soil samples, across 14 groups of Mean Annual Air Temperature, each containing approximately 40 samples. The orange line indicates the overall regression trend. Corresponding univariate regressions with AI are shown in Fig. S.5.

### 4.1.3 Complementarity of $MBT'_{5Me}$ and $MBT'_{6Me}$ to infer aridity

We propose to use the difference between $\mathrm{MBT}'_{5\mathrm{Me}}$ and $\mathrm{MBT}'_{6\mathrm{Me}}$ to track past aridity change, since it was shown that $\mathrm{MBT}'_{6\mathrm{Me}}$ variance is similar to that of $\mathrm{MBT}'_{5\mathrm{Me}}$ under humid conditions, while it is smaller under hyper-arid to dry sub-humid conditions

(Fig. 9A and B). This observation holds true in both soil and lacustrine samples. Therefore, following the trends of these two indices in the past could allow distinguishing between periods of constant humid conditions, whenever they are correlated with similar variance, and that of constant or shifted arid conditions (not correlated). Moreover, the current $\mathrm{MBT}'_{5\mathrm{Me}}$-AI correlation is negative (Fig. S.5A), while the $\mathrm{MBT}'_{6\mathrm{Me}}$-AI correlation is positive (Fig. S.5B), and similar to the soil water



content effect on $\mathrm{MBT'_{6Me}}$ (Dang et al., 2016b). Then, the $\Delta(\mathrm{MBT'_{5Me}}, \mathrm{MBT'_{6Me}})$ gives a quite reliable estimation of the
AI (i.e., the difference between the two indices, Fig. S.5C). This approach is more appropriate for soil than lake samples,
since the $\mathrm{MBT'_{6Me}}$ distribution in lacustrine samples is correlated with MAAT rather than AI (Fig. 6D and F), making the
$\mathrm{MBT'_{6Me}}$-based calibration more accurate than the $\mathrm{MBT'_{5Me}}$ one in ACA lakes.

The statistical independence between AI and MAAT has to be verified, especially since AI is related to temperature via the
Mean Annual Reference Evapotranspiration (cf. Eq. (3) and Trabucco and Zomer, 2018). Among ACADB, MAAT and AI are
independent bioclimatic variables ($R^2$ below 0.01 for lacustrine and about 0.15 for soil samples). Moreover, focusing only on
soil samples, the $\Delta(\mathrm{MBT'_{5Me}}, \mathrm{MBT'_{6Me}})$-AI relationship has been tested for similar MAAT values (Fig. 9C). The regressions
across MAAT classes are generally consistent, except at the extremes of the Aridity Index. The higher multiple $R^2$ compared to
the global model indicates that the relationship between $\Delta(\mathrm{MBT'5Me}, \mathrm{MBT'6Me})$ and AI is largely independent of MAAT.
When comparing this result with previous studies, we found similar $\mathrm{MBT'_{5Me}}$ and $\mathrm{MBT'_{6Me}}$ behaviour compared to MAAT
(correlations positive for $\mathrm{MBT'_{5Me}}$ and negative for $\mathrm{MBT'_{6Me}}$ in Guo et al., 2021). This is also supported by the evidence that
$\mathrm{MBT'_{6Me}}$ is correlated with AI and soil water content (Dang et al., 2016b; Guo et al., 2021). Although the difference between
ratios is still not commonly used for brGDGTs, this approach is increasingly applied to other geochemical proxies of past
climate parameters (Hällberg et al., 2024).

### 4.1.4 Controls of salinity on isomer ratios

Salinity is a major confounding factor in ACA, impacting both brGDGT distribution in the environmental space (Fig. 5B and E)
and the variance of FA and indices (Table 3). It is also significant that the salinity could be monitored through isomer content
(Fig. 8). Mainly, the $\mathrm{IR'_{6+7Me}}$ appears to be the more reliable index to track salinity gradients among lacustrine ACADB
samples, in line with Wang et al. (2021) and Kou et al. (2022). These studies have revealed the unusual over-representativeness
of 7-methyl compounds in brackish to hypersaline lacustrine lakes, like in the ACADB. Wang et al. (2021) also reported a
slight impact of salinity on $\mathrm{IR_{6Me}}$ which may be due to pH-salinity covariation in their database. Since pH and salinity are not
covarying in the ACADB soils, it explains why $\mathrm{IR_{6Me}}$ is less correlated with salinity for ACADB soils (Fig. 8A). However,
the two factors are covarying in the lacustrine dataset. This could explain why $\mathrm{IR_{6Me}}$-salinity regressions are similar between
soils and lacustrine samples ($R^2$ about 0.08) while $\mathrm{IR'_{6+7Me}}$ is not. We conclude that (1) the unusual over-representativeness
of 7-methyl due to salinity is important in lakes but not significant in soils, and (2), $\mathrm{IR'_{6+7Me}}$ is the more reliable brGDGT
index to track salinity changes in both sample types.

Especially for lakes, $\mathrm{IR_{6Me}}$ is well correlated with salinity for low salinity values (i.e., mainly fresh-water lakes, Fig. 8A and
Wang et al., 2021), while $\mathrm{IR_{7Me}}$ is more significant for higher salinity ranges (Fig. 8B). These thresholds of salinity classes are
attenuated when both 6- and 7-methyls are included in the ratio over 5-methyls (i.e., $\mathrm{IR_{6+7Me}}$ and $\mathrm{IR'_{6+7Me}}$ indices, Fig. 8C
and D). Mainly, lake salinity conditions may impact the *in situ* bacterial community responsible for the 6- and 7-methyl over-
abundances (Liang et al., 2024). The ACADB validates the use of $\mathrm{IR'_{6+7Me}}$ as a salinity proxy proposed by Wang et al. (2021),
in complement to previous proxies such as dinoflagellate cysts (Leroy et al., 2013), diatoms (Unkelbach et al., 2020), archaeol,
and caldarchaeol ecometric (Kou et al., 2022) or extended archeol (So et al., 2023).





### 4.1.5 Controls of pH on isomer ratios and cyclisation degree

Historically, brGDGT-based pH reconstructions were mainly conducted through cyclisation indices (e.g., CBT' and $\mathrm{CBT_{5Me}}$,
De Jonge et al., 2014). However, more and more studies highlight that the isomer ratio is also well correlated with pH changes
(Dang et al., 2016a; De Jonge et al., 2024a). From the ACADB, it appears that pH impacts the isomers more than the cyclisation
content (i.e., $\mathrm{IR_{6Me}}$ seems to be a more reliable proxy for pH than CBT' in ACA, Figs. 7 and 8). This is consistent with the
Inner Mongolian aridity transect study (Guo et al., 2021). At the global scale, CBT' presents a slightly higher correlation with
pH than $\mathrm{IR_{6Me}}$, but the sample type effect is stronger on CBT' (i.e., calibration for different sample types are more different
in slope and intercept for CBT' than $\mathrm{IR_{6Me}}$, Raberg et al., 2022a). $\mathrm{IR_{6Me}}$ may be more robust to infer past pH in a context of
shifting sedimentary flux. Particularly, panels (A) and (B) from Fig. 7, show that the cyclisation indices suffer from a correlation
disruption for alkaline soils. Here, a threshold appears after pH > 7.3 (threshold found by sensitivity analysis, Fig. S.2), in line
with a pH > 7.5 threshold demonstrated in Guo et al. (2021). At the global Chinese soils scale, CBT' correlation is very strong
with pH but not with MAP (Wang et al., 2019). It may show that the alkalinity effect on cyclisation is not enhanced by soil
aridity but by other phenomena. In Guo et al. (2021) where pH and aridity are associated, the more arid conditions do not relate
to increasing cyclisation number. Our results are in line with the study of Guo et al. (2021) who supports the use of $\mathrm{IR_{6Me}}$
for pH reconstruction. This consideration is important to keep in mind mainly for past brGDGT-based reconstructions carried
out in shallow lake (with important soil influx) and loess-palaeosol sequences (Lin et al., 2024), since the effect on lacustrine
samples is still unclear.

### 4.2 Assessing confounding factors combined effects

Although temperature, both MAAT and MAF, remains a major bioclimate parameter controlling the brGDGT distribution, we
have shown that other confounding factors such as aridity and salinity are at least as important in explaining the brGDGT dis-
tribution from ACADB. Particularly, confounding factors do not only impact their related indices used as proxies (e.g., MAAT
with $\mathrm{MBT'_{5Me}}$, salinity with $\mathrm{IR'_{6+7Me}}$, etc.) but also other indices (e.g., the ACA $\mathrm{MBT'_{5Me}}$ is also impacted by salinity). How-
ever, these confounding factors are typically studied independently, while in soil and lacustrine systems, multiple interacting
factors complicate the understanding of their combined effects.

### 4.2.1 Combined effect of pH with other confounding factors

Several complex interactions drive the pH effects on brGDGT assemblages. For example, pH is more related to the soil organic
matter content than salinity in arid contexts (Muhammad et al., 2008), and in ACADB there is no correlation between pH
and salinity for soil samples (Fig. 5). Similarly, humid environments are more likely to have organic-rich soil, influencing the
pH (Liang et al., 2019). However, isomer ratios are influenced by both these physicochemical soil properties. The alkalinity
interaction with aridity has been already reported from Chinese brGDGT soil studies (Yang et al., 2014; Dang et al., 2016b).
They show that the major brGDGT compounds are more diverse in arid than humid soils, implicating different correlations
between brGDGT indices, pH, and MAAT for dry and for wet soils (Wang et al., 2019). Although the brGDGT cyclisation





response to pH is globally well constrained, it is not the case in Chinese soils (Wang et al., 2019), but it is the case in northern Iran (Duan et al., 2022). Chen et al. (2021) introduced the use of soil water content as an intermediate parameter to clarify the pH-aridity interaction impact on brGDGT distribution. As a consequence, the brGDGT-based climate reconstructions of past archives in ACA need to be interpreted differently for alkaline samples.

### 4.2.2 Salinity effect and its relationship with aridity and sample types

The Procrustes rotation analysis performed on the ACADB reveals a different control of the environment on the brGDGT distribution from soil and from lacustrine origin. Particularly, in Fig. 5D and E, the salinity and AI were associated for lacustrine samples (i.e., the more humid the climate, the fresher the water), but surprisingly we observe the opposite association for soils. This could be due to the textural properties of soils, including the fact that salinization is more likely in clay than in sand (Muhammad et al., 2008). In this context, the numerous sandy desert samples in the ACADB from hyper-arid conditions do 560 not have high salt content. For ACADB lacustrine samples, aridity enhances the salt water content, which is consistent with actual observations (Williams, 1999) as well as during the Holocene (So et al., 2023). The salinity effect on not recalibrated temperature reconstruction may result in a temperature over estimation of more than 2 °C (Liang et al., 2024). Although the salinity effect on 7-methyl compounds is more and more understood for lacustrine samples (Wang et al., 2021; Kou et al., 2022; So et al., 2023; Liang et al., 2024), it remains understudied for soil samples. We have shown that the over-representativeness of 565 7-methyl brGDGTs is higher for lacustrine than soil samples. In soils, primarily the 6-methyl rather than the 5-methyl isomers seems to react to salinity, but this could be due to the combined effect of aridity, pH, and salinity. Salinity inferred by TDS is a bulk physicochemical parameter, and therefore, more details about the soil ionic composition are needed to refine the understanding of the salinity confounding factor effect (Chen et al., 2022; De Jonge et al., 2024a).

### 4.2.3 Combined effects of climate aridity on soil moisture, pH, and its consequences on the brGDGT distribution

In the ACADB, it appears that aridity enhances the abundance of cyclised and 6-methyl compounds over Ia, IIa, and IIIa (Figs. 7 and 8). This effect of aridity on both isomerisation (i.e., favouring 6- over 5-methyls) and cyclisation (favouring compounds with high internal cyclisation number) is a well-known effect in Arid Central Asia, showing the complex interaction between arid climate conditions, soil moisture, and pH (Dang et al., 2016b; Chen et al., 2021). Since the aridity effect on brGDGT is not directly settled by precipitation (Wang et al., 2019) but rather by the soil water content (Sun et al., 2019; Chen et al., 2021), this 575 could explain why the AI correlation with $IR_{6Me}$ and CBT' is not as strong for lakes as it is for soils (in which aridity directly impacts soil water content). Soil water content shows a clear impact on arid environments from both Chinese (Dang et al., 2016b) and African (Loomis et al., 2011) soils. Since the soil water content is not a limiting factor for the bacterial community in soil, the diminution of oxygen content in soil may be responsible (Li et al., 2018). Liang et al. (2019) suggest that this effect is particularly important in hyper- to semi-arid environments where bare soils (without vegetation cover) are dominant. 580 In contrast, in humid environments, the important vegetation cover changes the soil organic content and, by extension, the soil pH. These physicochemical causal links could explain the aridity's combined effects on ACA brGDGT distribution.





### 4.2.4 Toward a scale of confounding factors strength

Although several confounding factors have been identified worldwide (Naafs et al., 2017a; Dearing Crampton-Flood et al., 2020; Raberg et al., 2022a), it seems that the dominance order of these biases is study dependent. At a global scale, temperature
is the primary factor, followed by pH, independent of the type of sample studied (Raberg et al., 2022a). However, the linear relationship between brGDGT indices and both MAAT and pH is sample type dependent (i.e., the linear slope and intercept are not similar for soil, peat and lacustrine samples, Naafs et al., 2017a, b). Then, specific biases appear for specific sample types: mainly seasonality is important for lacustrine calibrations (Dang et al., 2018; Martínez-Sosa et al., 2021; Raberg et al., 2021). Even if seasonality also impacts soil-based calibrations (Deng et al., 2016; Dearing Crampton-Flood et al., 2020), it has
been shown that the monthly temperature fluctuations reported by brGDGT indices were similar to the average temperature (Cao et al., 2018), mainly due to the slow turnover of brGDGT production and deposit in soil samples (Weijers et al., 2011). In any case, seasonality is presented as one of the major confounding factors for global soil calibration (Dearing Crampton-Flood et al., 2020), while the aridity effect seems to be limited using Bayesian calibration. In brief, confounding factors studied at the global scale are slightly different from ones studied at regional and local scales.

At the Chinese soil scale, pH is the dominant controlling factor on brGDGT distribution (even before temperature, Wang et al., 2019), while focused on hyper- to semi-arid environments, the annual precipitation and aridity become more important than pH and temperature (Duan et al., 2022). In their study, all soil samples from northern Iran are alkaline and characterized by high temperature; then the environmental gradient is controlled by the precipitation gradient instead of the pH or seasonality. In Iran, the extreme alkalinity conditions should be responsible for the difference in confounding factor dominance with global
brGDGT studies. The seasonality bias on temperature is also climate dependent, with a stronger effect on areas with important seasonal variations, such as wide elevation gradients (Deng et al., 2016), continental areas, (Lei et al., 2016) and high latitudes (Dang et al., 2018; Cao et al., 2018). In contrast, seasonality effects are not reported from tropical area (Pérez-Angel et al., 2020; Häggi et al., 2023). From lacustrine samples from the Tibetan Plateau, salinity is the first confounding factor (Liang et al., 2024). The difference in site-specific confounding factors is supporting the idea that community shift is more determinant than
physiological plasticity to explain the brGDGT response to environmental parameters (Guo et al., 2021). This idea is also supported by the observed threshold in the brGDGT calibration under different temperature conditions (De Jonge et al., 2019). However, physiological plasticity remains supported by simulations and incubation experiments (Naafs et al., 2021; Halamka et al., 2023). These considerations give another valuable argument to assess the local or regional scale of the confounding factor effect on brGDGT index applicability.

## 4.3 brGDGT-based climate calibrations for drylands

We have shown that brGDGT-based indices are impacted in ACA by several confounding factors and their complex interactions. Here, we discuss the possibility of applying and developing specific calibration for drylands, mainly focusing on specific temperature and precipitation calibrations for specific sample groups (e.g., fresh, hypersaline, etc.).





### 4.3.1 Aridity and precipitation calibrations

The particular arid conditions in ACA have constrained several studies to propose brGDGT-based precipitation reconstructions (mainly MAP) in parallel or instead of MAAT calibrations (Dugerdil et al., 2021a; Duan et al., 2022). In both cases, the statistical relationship between $\mathrm{MBT}'_{5Me}$ or $\mathrm{MBT}'_{6Me}$ matched precipitation better than temperature, and MR calibrations based on brGDGT fractional abundances were proposed. However, precipitation was inferred by MR calibration or cyclisation indices rather than by methylation indices (Dugerdil et al., 2021a; Duan et al., 2022). Moreover, even if no proper calibration

is given, Lin et al. (2024) have shown a stronger linear correlation for $\mathrm{MBT}'_{5Me}$ with MAP than MAAT, especially for Chinese arid soils, giving specific interpretation for brGDGT loess-palaeosol sequences. A similar strong correlation with soil water content was also found in arid soils (Dang et al., 2016b).

### 4.3.2 brGDGT-temperature calibration

Our multivariate (Fig. 5) and univariate (Figs. 6 and 10) results have shown that several confounding factors can at least influ-
ence or at worst reverse the palaeothermometer calibration trends in drylands. The results of analyses of variance particularly evidence the influence of salinity and sample type on brGDGT FA. Mixing soil and lacustrine samples for such calibration raises the risk of misleading correlations. In ACA, $\mathrm{MBT}'_{5Me}$-based palaeothermometer should be applied carefully or, even better, it should be recalibrated by confounding factor classes. Mainly, the ACADB analysis of variance carried out on several brGDGT indices reveals that the *sample type* effect is weaker than salinity and as strong as pH. If the *sample type* effect is

already leading to particular calibration (i.e., soil, peat, and lacustrine ones) in global (De Jonge et al., 2014; Naafs et al., 2017a, b; Dearing Crampton-Flood et al., 2020; Martínez-Sosa et al., 2021) and regional calibrations (Sun et al., 2011; Yang et al., 2014; Chen et al., 2021), leading to specific peat, lake, and soil calibrations, caution needs to be taken with sample type calibrations in drylands. This is because, first, the *sample type* information for each sample is not always an indisputable observation. Lacustrine samples can be undoubtedly determined only from a deep lake (which is rare in drylands). Most ACA

lakes from the lowland basin are temporary ponds, seasonally drained, while several soil samples come from *solonchak* (i.e., saline rangelands covered by halophytic vegetation and periodically flooded, Gintzburger, 2003). For high-elevation lakes, due to their low level of water, the amount of soil influx from water springs and aerial dust is important. They also commonly have a semi-peatland behaviour due to the hydrophytic vegetation colonisation from belt to lake centre in the context of shallow water level (Cromartie et al., 2020; Robles et al., 2022). Among the difficulties encountered in reconstructing accurate temperatures

with brGDGT, the temperature offset between global calibration and study site climate context appears. In most cases, MAAT or MAF inferred by brGDGT shows an important shift between actual MAAT and reconstructed MAAT for the lacustrine top cores (Martin et al., 2019; Dugerdil et al., 2021b; d'Oliveira et al., 2023). Using a local calibration (Dugerdil et al., 2021a) or locally recalibrated global calibration (Chen et al., 2021), generally reduces this offset. Comparing pollen-based and brGDGT-based temperatures also reveals that the brGDGT-MAAT relationship has wider temperature variation over the same time span

(Robles et al., 2022; d'Oliveira et al., 2023).





### 4.3.3 Confounding factors effect on temperature calibrations

While $\mathrm{MBT}'_{5\mathrm{Me}}$-based temperature calibrations are commonly specified only based on the sample type (i.e., separate calibrations exist for soil, peat, and lacustrine samples), we have shown previously that the statistically most influential confounding factor in the ACADB was salinity before pH and sample type. Mainly, the Procrustes rotation analyses show similar multivari-

ate spaces for soil and lacustrine samples (Fig. 5C and F). Based on this finding, specific calibrations based on salinity classes are tested on Fig. 10. The same tests were applied for pH (Fig. S.6), aridity (Fig. S.7) and sample type (Fig. S.8).

For the salinity classes, all except adjusted-regression hyposaline are improved compared to the global regression (i.e., the full ACADB), especially for fresh water, hypersaline and saline ($R^2_{\mathrm{adj}}$ of 0.61, 0.49, and 0.37, respectively). This method principally improves the statistical result for extreme classes. However, this improvement could be artificial and only produced

by statistical biases such as the dataset size or the reduction of variance among groups. To ensure relevant salinity-specific calibrations, the $z$-statistic checks differences between regressions. Fresh calibration has a different slope than hyposaline and the whole dataset with z(a) = 2.7*** and 2.5***. Saline samples share a similar slope with hyposaline but a different offset with z(b) = -2.2**. No conclusion can be made for hypersaline due to high variance. Saline and likely hypersaline have a similar

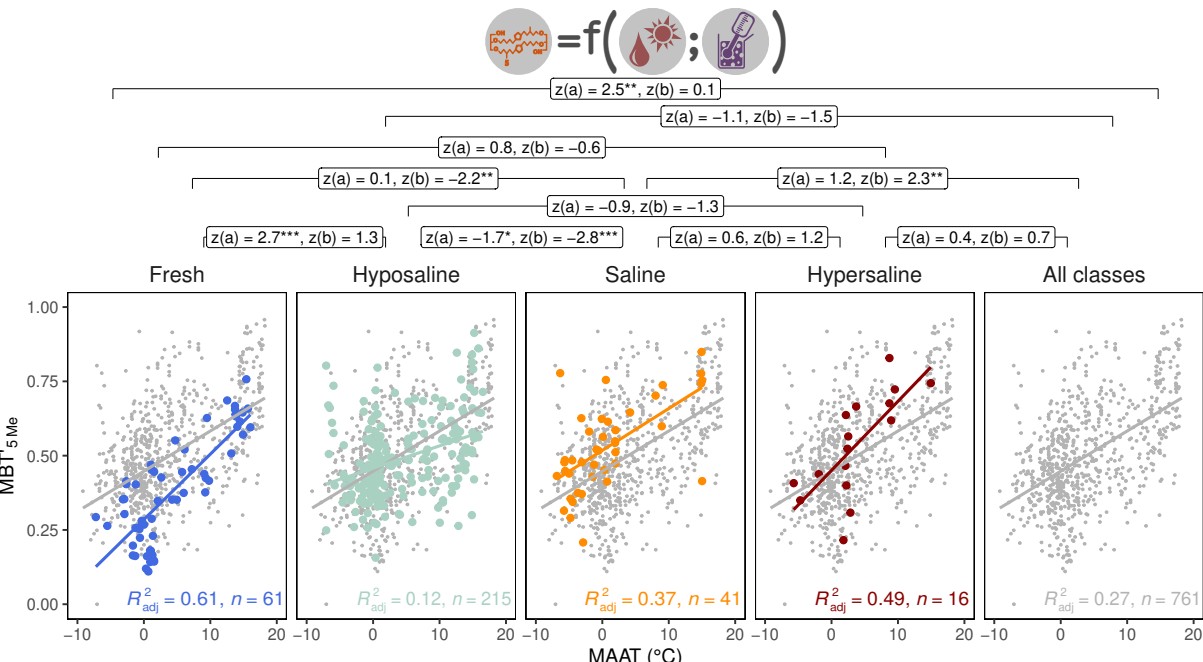

**Figure 10.** Effect of salinity (i.e., the TDS of each surface sample) on the linear relationship between temperature (here MAAT) and the $\mathrm{MBT}'_{5\mathrm{Me}}$. The analysis is based on the full ACADB. Based on the salinity classes (i.e., fresh, hyposaline, saline, and hypersaline), different temperature calibrations based on the degree of methylation with MAAT = a $\times$ $\mathrm{MBT}'_{5\mathrm{Me}}$ + b are proposed. Using the $z$-statistic with its $p$-value (Clogg et al., 1995), the significance of the difference between the slopes (a) of the linear regression is evaluated. Similarly, the $z$-statistic was used for the intercept (b) differences. For the $p$-values we have *** for p $\leq$ 0.01, ** for p $\leq$ 0.05, and * for p $\leq$ 0.1.





slope to hyposaline but a significantly higher offset (z(b) = -2.8*** with hyposaline, z(b) = 2.3*** with the whole dataset).
This analysis allows for ACA $\mathrm{MBT}'_{5Me}$ calibration based on salinity classes [Eq. (6)].

$$\mathrm{MAAT}_{\mathrm{Fresh}} = -6.37 + 28.03 \times \mathrm{MBT}'_{5Me}, \ (n = 61, R^2_{\mathrm{adj}} = 0.61, \mathrm{RMSE} = 4)$$

$$\mathrm{MAAT}_{\mathrm{Sal.}} = -11.1 + 23.17 \times \mathrm{MBT}'_{5Me} (n = 45, R^2_{\mathrm{adj}} = 0.32, \mathrm{RMSE} = 5.22) \tag{6}$$

$$\mathrm{MAAT}_{\mathrm{Hypersal.}} = -8.33 + 22.36 \times \mathrm{MBT}'_{5Me} (n = 17, R^2_{\mathrm{adj}} = 0.47, \mathrm{RMSE} = 3.88)$$

### 4.4 Recommendations for brGDGT applications to past records in drylands

#### 4.4.1 Differential brGDGT sources effect in the past

If one of the solutions to reduce biases for brGDGT-based climate reconstructions in the past is to lead *sediment type* particular
brGDGT calibrations, some tricky issues remain. (1) The sediment type characterisation has to be reliable. Several studies attempt to use brGDGT themselves as a proxy of brGDGT sources based on mixing models (Martin et al., 2019) or classification machine learning approaches (Martínez-Sosa et al., 2023; Cromartie et al., 2025). Other proxies of the sample type could be a more reliable solution to evaluate the environmental condition of brGDGT deposit (Robles et al., 2022; d'Oliveira et al., 2023). (2) The brGDGT influx is not always related to the sediment deposit flux itself. In some particular conditions, a lake can record
soil-produced brGDGT coming from the watershed instead of *in-situ* produced brGDGT (Zhao et al., 2021; Robles et al., 2022). In some other conditions, the *in-situ* production is dominant in the brGDGT assemblage (Wang et al., 2021; Kou et al., 2022). (3) When sediment types vary along a core, accurately characterizing them becomes challenging, making it difficult to apply a single brGDGT-based climate calibration. For instance, using a uniform lacustrine calibration may yield inconsistent results when applied to both hypersaline and freshwater lake samples, as illustrated in Fig. 10.

#### 4.4.2 Confounding impacts and correction in the past

Then, the past sediment type characterisation appears to be insufficient to reliably apply selective brGDGT-based climate calibrations. The main confounding factor effect has to be taken into account. However, these confounding factors are study-context dependent: the saturation effect of the $\mathrm{MBT}'_{5Me}$ and the vegetation buffer in tropical areas (Pérez-Angel et al., 2020; Häggi et al., 2023), the important seasonality in the Arctic (Raberg et al., 2021), soil moisture and salinity impact in drylands
(Fig. 10; Dang et al., 2016b; Kou et al., 2022), etc. A second limitation of this grouping factor selective calibration approach in the past is that the confounding factor impact is not always stationary over time. For instance, salinity (So et al., 2023) or vegetation cover (Robles et al., 2022; d'Oliveira et al., 2023) dramatically shifted in the past. It is particularly the case during the Holocene in ACA (Chen et al., 2024). In this case, inferring confounding factor covariation has to be considered. Several studies attempt to track these covariations using brGDGT-based confounding factor proxies, such as isomer ratios (Liang et al.,
2024). Some others use a multi-proxy approach (e.g., pollen or chironomids) to independently infer the confounding factor variation trends (Dugerdil et al., 2021a, b; Robles et al., 2022; d'Oliveira et al., 2023).

To improve the regression between $\mathrm{MBT}'5Me$ and MAAT, we apply calibrations using different grouping factors, mainly salinity (Wang et al., 2021, Fig. 10), but also pH and aridity (Figs. S.6 and S.7). Chen et al. (2021) split the GDGT dataset



by surface sample pH (threshold of pH = 7), which is reflected in the ACADB CBT'-pH relationship (Fig. 7A and B) and
reported in arid soils (Guo et al., 2021). Similarly, Wang et al. (2021) improved the $\mathrm{MBT'5Me}$-based temperature calibration
for Tibetan lacustrine samples by including salinity, but quantitative values for these factors are unavailable for past brGDGT
sequences.

One of the solutions to fulfil this lack of information about the deposit system in the past is to make differential calibrations
not on confounding factor values but on brGDGT-based proxies of these confounding factor values. Such an approach has been
tested in Véquaud et al. (2022), using the Community Index (De Jonge et al., 2019) with mitigated regression improvement.
In Véquaud et al. (2022), the two clusters based on the Community Index threshold were thought to improve the temperature
reconstruction by showing specific regression for each of them. However, the two sub-groups give mitigated determination
coefficient ($R^2$ of 0.20 and 0.71). More recently, Liang et al. (2024) propose to use the isomer ratio (since this index includes
both salinity and pH effects) to constrain $\mathrm{MBT'_{5Me}}$-based MAAT calibration with more reliable results in saline lakes. All these
attempts are still incomplete to totally correct the brGDGT-based temperature reconstruction bias, even if they significantly
reduce the over-estimation of MAAT. In any case, a careful examination of the brGDGT distribution from past archives is
essential to minimize errors resulting from the inappropriate selection of applied calibrations.

## 5 Conclusions

The brGDGT-based palaeothermometer is one of the most promising approaches to improve our understanding of past climate
in different regions of the world. However, based on the comparison between an ACA-centred database and the world surface
sample database, our study has shown that:

1. Drylands suffer from particular climate and physicochemical properties of soils and lakes, enhancing the impact of
   confounding factors on brGDGT-based $\mathrm{MBT'_{5Me}}$ and fractional abundances' relationship with MAAT.

2. Among the confounding factors (i.e., pH, aridity, salinity, and sample type), salinity is the most dominant, followed by
sample type and pH. However, aridity plays a major role in the brGDGT variance among the dataset. Moreover, these
   biases cannot be studied individually since their interactions are not always similar. For instance, the salinity control on
   brGDGT isomerisation is different in soil and lacustrine deposit contexts.

3. In order to use brGDGT as a proxy for sediment physicochemical conditions, it appears that the $\mathrm{IR'_{6+7Me}}$ is the best
   index of salinity, while $\mathrm{IR_{6Me}}$ is the best for pH reconstruction despite its saturation effect for pH < 4 and pH > 10. For
aridity, drawing $\Delta(\mathrm{MBT'_{5Me}}, \mathrm{MBT'_{6Me}})$ gives a fairly reliable estimate.

4. $\mathrm{MBT'_{5Me}}$ relationship with MAAT is very limited in ACA, especially for lacustrine samples, mitigating the applicability
   of palaeothermometers based on methylation indices. However, the specific sub-calibrations for different environmental
   classes (mainly salinity and aridity classes) dramatically improve the linear regression strength. This report paves the
   way for a specific calibration application on past brGDGT sequences based on environmental classes inferred or by
brGDGT indices or by other independent proxies (e.g., pollen or chironomids).



Mainly, even if the brGDGT signal in drylands such as ACA is mitigated and the number of confounding factors is sometimes difficult to unravel, it remains a very promising tool to improve our understanding of both past climate and future forecasting (Tierney et al., 2020). However, some work still remains in process, including the increase of arid sites in the database (both for the calibration process associated with exhaustive physicochemical and bioclimatic properties of the samples and for the past brGDGT sequences enhanced by a multi-proxy approach) and the development of a machine learning approach, which promises a more powerful unravelling process of confounding factor comprehension.

*Code availability.* The R scripts used to clean, analysed the ACADB surface brGDGT data and plot all the figures presented in this study (except Fig. 1 and 2 realized in `QGIS` and `Inkscape`) are open and available on the following GitHub repository: https://github.com/ LucasDugerdil/GDGT_ACADB/.

*Data availability.* Sample location and features, brGDGT fractional abundances and concentrations, brGDGT-based indices, bioclimate, and physicochemical parameters are available on PANGAEA dataset XXXX.

*Author contributions.* *Conceptualization*: LD, SJ, OP, MG; *Data curation*: LD; *Formal analysis*: LD, AC, GM, SAA; *Funding acquisition*: SJ, XH, BB, FC, GM; *Investigation*: LD, SB, XH, JA, RS, IM, SI, SM, EA, SI, PG; *Methodology*: LD, SJ, OP, MG; *Project administration*: LD, SJ, OP, MG; *Resources*: LD, SB, XH, FC, DE, JA, BB, JH, LS, KM, RS, IM, SI, SM, EA, SI, PG; *Software*: LD, AC; *Supervision*: LD, SJ, OP, MG; *Validation*: LD, SJ, OP, MG, XH, FC, BB; *Visualization*: LD; *Writing – original draft*: LD; *Writing – review & editing*: all authors.

*Competing interests.* The authors declare that they have no known competing financial interests or personal relationships that could have appeared to influence the work reported in this paper.

*Acknowledgements.* This research was funded, in whole or in part, by ANR, Grant ANR-22-CE27-0018-02, and Grant ANR-20-FRAL-0006. The ENS de Lyon financed the PhD scholarship of L. Dugerdil. We are grateful to *La Tendresse*, Sarah Millet-Amrani, and Zoé Hernandez for their valuable support in France; Nozimbek Namozovich and Sokhib Abdusamatov in Uzbekistan; and Gulmirzozoda Masnavi Gulmirzo in Tajikistan. We are grateful to Vivien Mai Yung Sen for his help in high mountain sampling in Hissar and Pamir. Thanks to Elodie Brisset (IMBE) and Johanna Lhuillier (Archéorient) for their support during the Uzbekistan field trips. Bazartseren Boldgiv is supported by the Taylor Family-Asia Foundation Endowed Chair in Ecology and Conservation Biology. This is an ISEM contribution XXXX.



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
