# Peer review of "BrGDGT-based palaeothermometer in drylands: the necessity to constrain aridity and salinity as confounding factors to ensure the robustness of calibrations."

_EGUsphere, 2025_

## Referee Comment (RC2)

**Review of Dugerdil *et al.* for *Biogeosciences***

**Joseph B. Novak**

**Summary**

Dugerdil *et al.* present an analysis of soil and lacustrine brGDGT samples from Central Asia that considers new and published data from the region. The analysis focuses on the role of salinity, pH, and sample type in obscuring the relationship between the degree of brGDGT methylation and temperature. The analysis indicates that the generally weak relationship between the $MBT'_{5Me}$ index and mean annual air temperature can be attributed to these variables, to different degrees. The work presented here is an important step forward for understanding the systematics of brGDGTs in Central Asia and I look forward to the publication of this work once my concerns are addressed.

Please find my specific concerns about the manuscript below.

Warm regards,

- Joseph Novak

**Major Comments**

**Abstract (L4–6):** This study does not introduce the Arid Central Asian brGDGT database, as this same dataset was analyzed in a recent publication by this group in *Paleoceanography and Paleoclimatology* (https://doi.org/10.1029/2025PA005214). Same applies for the statement in **Lines 93–94**. I think it would be more appropriate to discuss this work as a further exploration of the Arid Central Asian brGDGT database.

**"confounding factors":** Throughout the text, the phrase "confounding factors" is used to refer to any variable other than temperature. I think a slight reframing of the text would be useful to clarify what is of interest here. Specifically, salinity and pH as they relate to brGDGT distributions and aridity. Discussing these factors as independent variables of interest rather than confounding factors may be a useful way to highlight the importance of the work presented here.

**Figure 1 and database composition:** There are some inconsistencies in the Figure that make me question whether database is entirely composed of sediment/soil samples. Firstly, there are purple stars plotted on the map, but these are not shown in the Figure legend. In the case of the Lake Baikal basin (where I work), these mark the locations of moss polster brGDGT samples reported in previous work by Dugerdil et al. (2021). Similarly, the blue stars within Lake Baikal appear to mark suspended particulate matter samples reported by (De Jonge et al., 2015; note that this paper is not cited by the authors). Neither of these data types are soil / sediment. Based on these observations in the region I am familiar with, I question whether the rest of the database was constructed carefully and ask that the authors carefully screen the data or more clearly distinguish between sample types in this figure and in their analysis, particularly if different types of lacustrine (sediment vs. suspended particulate matter) and terrestrial (soil vs. plant tissues) samples are being considered in combined datasets. This is especially important to distinguish since the text discussing the new data presented in this study communicates a much stricter definition of a lacustrine or soil sample than what appears to be the rule in the portion of the database composed of previously published data.

On a related note to the comment above, I am doubtful that Lake Baikal is a useful point of comparison to the other samples in the ACA database for generating a regional calibration since this lake is remarkably different from the much smaller / ephemeral lakes in the rest of the dataset. This may merit a bit of text to talk about if you want to include data from Lake Baikal in this analysis.

**L119–132:** This section would benefit from some supplementary figures showing the environmental variables you are discussing, either as maps or histograms. As it stands, this is a pretty dense wall of text to get through without any visualization to help the reader understand what is being communicated.

**L142–143:** "The chemical characteristics of the ACA surface samples include pH and Electro-Conductivity (EC, Fig. 2, step 2), both measured *ex-situ* in the laboratory, even for lacustrine samples." Was there anything done to make sure that microbial respiration post sample collection did not cause the samples to become more acidic during transport? For the lacustrine samples, what is the logic that the pH or salinity of the pore water in the sediments is equivalent to the properties of the lake water? This is particularly concerning for pH, since the pH of sediment pore waters can change substantially (and therefore become different from the overlying water column) in even the upper few cm of the sediments (e.g., Bachmann et al., 2001; https://doi.org/10.1016/S0375-6742(01)00189-3).

**3.1.1 brGDGT concentrations (L262–269):** Comparing concentration measurements across studies is really tricky because there is no authentic brGDGT standard. The $C_{46}$ internal standard (Huguet et al., 2006) is useful for considering relative differences in concentration between samples run at the same time, but this method does not control for instrument drift (see Figure 4 of Huguet et al., 2006). Some work needs to be done to show whether the uncertainties introduced by this shortcoming of the $C_{46}$ method pose a problem for the analysis here. Or, at the very least, some text needs to be added to acknowledge this shortcoming of the concentration data. The discussion of the concentration data here feels somewhat random since they do not appear to be discussed later in the text. If they are going to be brought up, it would be interesting to understand if concentration is related to any of the indices that describe the brGDGT distributions. For example, we found a relationship between brGDGT concentration and $IR_{6Me}$ in lacustrine sediment samples from Lake Baikal (Novak et al., 2025).

**4.2.4 Toward a scale of confounding factor strength (L582–609):** This section does not present any sort of quantitation of the strength of the confounding factors in the temperature-methylation relationship, so the name of this subsection is not appropriate. Rather, this text is a discussion of previous work on the topic. Is there maybe a way to tie this discussion in with the previous sections a bit more coherently? As it stands, I struggled to understand what the takeaway message of these two paragraphs is.

**4.3.3 Confounding factors effect on temperature calibrations (L646–660):** Text needs to be added to the methods that explains how lakes were classified into the different "salinity classes." Specifically, the cutoff values for the different classes need to be given and, if they are not taken from another publication, justification for these cutoffs should also be provided.

**Figure 10:** This figure design is really confusing to me. What is going on with the z-scores at the top of the figure? What do the boxes with lines represent?

**Equation 6:** This should probably be broken up into sub-equations (i.e., 6.1, 6.2, etc.) since there are multiple equations listed here. Also, five regressions are shown in Figure 10, but only three equations are given here. Why? This should be justified in the text or the other two equations provided.

**Minor Comments**

**L26:** I do not think we can really call brGDGTs a "new" proxy anymore.

**L29:** "…well preserved *in lake sediments*…" current wording is not grammatical.

**L30–33:** This sentence is written in a confusing way that is not understandable to someone who is not familiar with the brGDGT proxy.

**L33:** "The relationship is clear and linear." I think there are missing words here. It is not clear what relationship you are talking about.

**L34:** This sentence would benefit from consolidation.

**L35:** "*acidic* lakes"

**L38:** This statement could use a citation or two as an example.

**L55:** "pentamethyls" is a bit informal. Here and elsewhere this language should be replaced with "pentamethylated brGDGTs," adapted as appropriate for the molecules discussed in each instance.

**L60:** "This leads to a *different* temperature relationship between brGDGTs *in soils* and MAAT…"

**L68–69:** "…low organic matter **content**…"

**L85:** "Salinity is *thought* to influence…" is probably more appropriate since there is not (at least to my knowledge) a biochemical explanation for why the salinity would cause preferential synthesis of different brGDGTs.

**L89:** isn't MLR the typical abbreviation for multiple linear regression?

**L95:** "…totaling 761 *samples*."

**L97:** What specific temperature variable(s) are you correlating to?

**L112:** What does "the data location was randomly selected" mean?

**L138:** *core tops* rather than "top cores"

**L193–197:** The justification for not looking at DC or DC' does not make a lot of sense to me. Should not all of the indices be considered?

**L256:** "the size of the data is high" this reads a bit awkwardly. How about, "since *the number of samples is large*."

**2.5 Database compilations:** If you are going to include Lake Baikal data, the database is missing recently published surface sediment samples from Lake Baikal (Novak et al., 2025).

**Figure 3:** Some text is needed in the figure caption discussing how the bounds of the box and whisker plots are defined.

**3.1.2 brGDGT fractional abundances:** More figure callouts are needed in this section. Figure 3 is rather complicated, so it would be helpful to guide the reader through each of the panels.

**L276:** "The 7-methyl isomers *are more abundant* in lacustrine…"

**L277:** If you are talking about Figure 3, these are box and whisker plots, not histograms.

**L256–282:** The language here needs to be tightened up. When talking about the data, "trends" would imply some sort of regression exercise that generated an estimate of slope. Here, it is more important to talk about differences in the median values in the sample sets separated by aridity class (which is what the box and whisker plots are showing). This could be coupled with significance tests in differences of the mean.

**L292–293:** What do you mean by the aridity effect? This is not clear from the text.

**L304:** I would not say this is a "clear aridity gradient" since there is substantial overlap between the distributions plotted on the upper axes of the plots in Figure 5. This result is, perhaps, suggestive of an aridity gradient.

**3.2.1:** Some text explaining why this analysis was done and what the VIF value means would be really useful to guide the reader through this dense section of the text.

**L341–345:** Figure callouts are needed here.

**L364:** The meaning of the "*" needs to be explained, or the p-value can just be written as $p << 0.001$ or however else is appropriate for each relationship.

**L361–372:** Is there a figure associated with these tests and statements (presumably Figure 6)? Callouts are needed.

**L372:** Is there a figure or statistical test (preferably both) you can point to that indicates that "hyper-arid samples often have extreme values" ?

**L384–386:** I think you mean "…there is a strong *correlation* between…"

**L428–429:** The confounding factor in relation to what? Estimating MAAT from one of the methylation indices?

**L454:** "attenuation" is not the right word to use. *Weakening* is probably what you mean.

**L475:** typo here.

**L500:** Again, a confounding factor for what?

**L512–514:** "These thresholds of salinity classes are attenuated…" I do not really understand what you mean here. What thresholds do you mean? I do not see any in the Figure that is referenced. I understand that you are talking about the difference in the correlation strength to salinity of $IR_{6Me}$ vs. $IR_{6+7Me}$, but the language here is hard to follow.

**L526–527:** What is a "correlation disruption"?

**L528:** "global Chinese soils" seems self-contradictory.

**L562:** What is a "not recalibrated temperature reconstruction" ?

**L564:** "*over-representation*" Also, not really sure what you mean here.

**L615:** "constrained" is not the correct word to use here. I think you mean "***led***" but you should make sure of that.

**L643:** What is a "locally recalibrated global calibration" ?

**L644-645:** Could some of this not be because of the typically different seasonality of brGDGT and pollen temperatures? As in, brGDGTs typically are a proxy of MAAT or MAF while pollen is typically mean temperature of the warmest month or mean temperature of the coldest month? Temperature change is expected to have a different magnitude across different seasons from climate model simulations and modern observations (Feldl and Merlis, 2021, is an example from the high latitudes).

**L684–685:** I do not think "lead" is communicating your intended meaning in this sentence, but I cannot figure out what you are trying to say so I cannot suggest a better word.

**L695:** Rather than "mitigated" I think you mean "***limited***."

---

## Author Comment (AC1)

**Dear Yan Bai,**

On behalf of all co-authors, I would like to warmly thank you for your thorough review and constructive feedback. Below, I address your main concern.

**Major concern 1 :**

The preprint was submitted to the EGU discussion platform for peer-review consideration in *Biogeosciences* on 28 July 2025, while the second article published in *Paleoceanography and Paleoclimatology* (https://doi.org/10.1029/2025PA005214) appeared later, on 6 October 2025. Initially, this preprint was intended as the first part of the project, focusing on the description of 162 new samples and the comparison of GDGT distributions across Arid Central Asia. The *Paleoceanography and Paleoclimatology* paper (*Dugerdil et al., 2025*) was conceived as the second part, dedicated to comparing traditional and machine learning-based calibration methods.

However, due to differences in the peer-review timelines between *Biogeosciences* (which includes a public discussion phase) and *Paleoceanography and Paleoclimatology* (which follows a classical closed review process), the second part was published before the first.

Although both manuscripts rely on the same dataset (162 new samples combined with 599 previously published ones), the current preprint presents, for the first time, a detailed description and analysis of the fractional abundances and major GDGT indices for these new samples. In contrast, Dugerdil et al. (2025, Paleo) uses the ACADB dataset solely to train machine learning models, without providing the complete data characterization presented here.

However, we agree that, in the current state of publication, it appears more accurate to remove the term "new" in the current study. This word has been removed along the manuscript and a reference to Dugerdil et al., (2025, Paleo) has been added in line 93-104 within the introduction with : "*This study relies on the first regional database of surface brGDGT samples for drylands, aiming to identify the key climate and environmental parameters influencing their distribution. This dataset, referred to as the Arid Central Asian brGDGT Surface Database (ACADB), includes brGDGT assemblages from various sites across the region, totalling 761 sites. This dataset was compiled by Dugerdil et al., (2025) to train machine learning models for climate reconstructions. The dataset combines 162 new samples collected across four ACA countries with 599 previously published records (Fig. 1). In Dugerdil et al. (2025), machine learning calibrations outperformed traditional linear models, suggesting that confounding factors weaken linear brGDGT–temperature relationships. The present study tests this hypothesis by analysing modern brGDGT distributions against key climate parameters, mainly aridity, temperature, and precipitation, as well as chemical characteristics such as pH, salinity, and sample type (soil or lacustrine). The results are then compared with the global Worldwide brGDGT Surface Database (WDB; modified from Raberg et al., 2022b) to assess whether similar brGDGT patterns are observed at both regional and global scales.*". Also the Materials and Method section have been modified as follow (l. 211-213) "*Two databases are compared in this study (Figs. 1 and 2, steps 4 and 5). The Arid Central Asian Data Base (ACADB, n = 761) gathers samples from Dugerdil et al. (2025) used to train machine learning calibrations, as well as samples collected from previously published studies, listed in Table 1 and Table S.2.*".

**Major concern 2 :**

We agree that previous studies have shown that linear regressions (i.e., MBT-based calibrations) and multiple linear regressions perform less effectively in drylands than in more humid environments. In the present study, we build upon these findings by testing a broader range of indices, environmental parameters, and an expanded dataset, the ACADB.

To further enhance palaeoclimate reconstructions in drylands, we propose two methodological improvements: (1) a salinity-specific set of MBT-based calibrations, and (2) a new index based on the difference between MBT'$_{5Me}$ and MBT'$_{6Me}$ to track aridity changes. These improvements directly result of the analyse of the modern brGDGT response to environmental gradients.

The detailed evaluation of quantitative reconstructions derived from modern brGDGT samples (i.e., proper climate calibration process) constitutes the primary focus of *Dugerdil et al. (2025, Paleo)*. We therefore consider the two publications to be distinct yet complementary, each addressing a different aspect of the overarching research objective: (1) the brGDGT distribution response against climate and soil properties, and (2) the quantitative climate calibration process.

However, to make this articles distinction and complementarity more clear we modified the present manuscript by adding a section *Temperature calibration errors* in section 4.3.1 from the Discussion. This section is supported by a new figure (Fig. 10) and the following text (l. 648-656): "*As a result of the multiple confounding factors influencing the response of brGDGTs to climate, and their complex combined effects, traditional temperature calibrations exhibit substantial errors when cross-validated using the ACADB dataset (Fig. 10). This is true for both the Mean Annual Air Temperature (Fig. 10A) and the mean air temperature of Months Above Freezing (Fig. 10B). At the ACADB scale, local calibrations show a significant average bias, producing either overly warm (Yang et al., 2014; Sun et al., 2011; Thomas et al., 2017) or overly cold estimates (Wang et al., 2016). Although global calibrations reduce this offset, they still display wide dispersion (ranging from –20 to 35 °C). These large errors persist across various statistical approaches, including quadratic and multiple linear regressions, MBT'$_{5Me}$-based, and Bayesian calibrations. Altogether, these findings highlight the need for developing dedicated calibrations for dryland environments, particularly focusing on temperature and precipitation reconstructions tailored to specific sample types (e.g., freshwater vs. hypersaline systems).*".

**Minor comments :**

Line 34: We changed "air" by replacing it with "ambient temperature".

Line 45: We changed "CBT'" to "CBT$_{5Me}$".

Line 55: The word "thus" have been changed by "For instance"

Table 2 and Line 192: The "CBT'$_{5Me}$" indication have been corrected to "CBT$_{5Me}$" all along the manuscript and in the figures as well.

---

## Author Comment (AC2)

**Dear Joseph B. Novak,**

On behalf of all co-authors, I would like to warmly thank you for your thorough review and constructive feedback. Below, I address your main concern.

**Major Comments**

**Abstract (L4–6):** This study does not introduce the Arid Central Asian brGDGT database, as this same dataset was analyzed in a recent publication by this group in Paleoceanography and Paleoclimatology (https://doi.org/10.1029/2025PA005214). Same applies for the statement in Lines 93–94. I think it would be more appropriate to discuss this work as a further exploration of the Arid Central Asian brGDGT database.

The current study was submitted in Biogeosciences prior to the study published in October 2025 in Paleoceanography and Paleoclimatology (Dugerdil et al., 2025). Both are complementary: the present study is thought to introduce the Arid Central Asian brGDGT database and present its response to environmental variables, while Dugerdil et al. (2025) is focusing on new statistical tools (mainly machine learning) to develop accurate brGDGT-based climate reconstruction trained from the ACADB. This explains why the terms "introduce" and "new" were present in the submitted manuscript and can now be adapted.

To make this point clearer, we modified in the abstract, the lines 4-6 with "*This study further explores the recently compiled Arid Central Asian (ACA) brGDGT surface Data Base, a regional dataset comprising 761 surface samples from the drylands of ACA.*".

Then, in the introduction, we modified the presentation of the approach carried in this study by (l. 93-102): "*This study relies on the first regional database of surface brGDGT samples for arid regions, with the objective of determining the principal climatic and environmental factors affecting their distribution. The Arid Central Asian brGDGT Surface Database (ACADB) comprises brGDGT assemblages from multiple locations within the region, totalling 761 samples. This dataset was compiled by Dugerdil et al. (2025) to train machine learning models for climate reconstructions. The dataset integrates 162 new samples gathered from four ACA countries with 599 previously published records (Fig. 1). Dugerdil et al. (2025) demonstrated that machine learning calibrations surpassed conventional linear models, indicating that confounding factors diminish the brGDGT–temperature correlations. This study evaluates the hypothesis by analysing contemporary brGDGT distributions in relation to critical climate parameters, including aridity, temperature (both Mean Annual Air Temperature, MAAT, and the seasonal mean temperature of Months Above Freezing, MAF), and precipitation, alongside chemical characteristics such as pH, salinity, and sample type (soil or lacustrine).*".

**"confounding factors":** Throughout the text, the phrase "confounding factors" is used to refer to any variable other than temperature. I think a slight reframing of the text would be useful to clarify what is of interest here. Specifically, salinity and pH as they relate to brGDGT distributions and aridity. Discussing these factors as independent variables of interest rather than confounding factors may be a useful way to highlight the importance of the work presented here.

The main aims of this study is to assess the reliability of the brGDGT to produce accurate temperature reconstructions. The question of the possibility to use brGDGTs in the past to track other factor variations, is secondary. That is why we mainly call these variables as "confounding factors", term widely used by the brGDGT community. However, we agree that we inappropriately

use this term in some context, especially when testing the second question. Then, throughout the manuscript we modified the text by using *controlling factors* or *environmental variables* when considering the parameters that may control the brGDGT distribution, and *confounding factors* when focusing on the other environmental variables that impact the relationship between a brGDGT index and its primary controlling factor. For instance, the MBT'$_{5Me}$-temperature relationship is also impacted by the salinity in lacustrine sediment, here the salinity is called as *confounding factor*. Some modification of the text may been founded on lines 248, 441, 457, 461, 527 and 615 for instance (see the track-changed version of the manuscript for more details).

**Figure 1 and database composition:** There are some inconsistencies in the Figure that make me question whether database is entirely composed of sediment/soil samples. Firstly, there are purple stars plotted on the map, but these are not shown in the Figure legend. In the case of the Lake Baikal basin (where I work), these mark the locations of moss polster brGDGT samples reported in previous work by Dugerdil et al. (2021). Similarly, the blue stars within Lake Baikal appear to mark suspended particulate matter samples reported by (De Jonge et al., 2015; note that this paper is not cited by the authors). Neither of these data types are soil / sediment. Based on these observations in the region I am familiar with, I question whether the rest of the database was constructed carefully and ask that the authors carefully screen the data or more clearly distinguish between sample types in this figure and in their analysis, particularly if different types of lacustrine (sediment vs. suspended particulate matter) and terrestrial (soil vs. plant tissues) samples are being considered in combined datasets. This is especially important to distinguish since the text discussing the new data presented in this study communicates a much stricter definition of a lacustrine or soil sample than what appears to be the rule in the portion of the database composed of previously published data. On a related note to the comment above, I am doubtful that Lake Baikal is a useful point of comparison to the other samples in the ACA database for generating a regional calibration since this lake is remarkably different from the much smaller / ephemeral lakes in the rest of the dataset. This may merit a bit of text to talk about if you want to include data from Lake Baikal in this analysis.

We want to acknowledge you for this very careful check-up of the data. Actually, you found and highlight two errors in our data set process.

First, about the Figure 1 the purple symbol was actually a mistake. This symbol displayed the *moss polster samples* analysed by our research group (In Armenia, Mongolia and the Baikal basin). This sample sub-type is among the *soil* samples. We modified the Figure 1 accordingly, by changing these purple symbols into red symbols.

Second, you are true that some *lacustrine* samples are *SPM lacustrine* samples. After a careful and complete check-up of the dataset process, we found that 8 samples were SPM. Mainly the ones from the study of De Jonge et al. (2015). The paper were not cited because our data process were supposed to removed these SPM samples from the dataset. Actually, a mistake in the R routine was responsibles for this mistake. To correct that, the 8 SPMs were fully removed from our study. The ACADB size changed from 761 to 753, including 193 lacustrine samples instead of 201. We corrected the Table 1 and Figure 1 by removing these 8 Lake Baikal SPM data.

Afterward, all the data analyses and figures were re-run in order to homogenise the study with this cleaned version of the dataset. The results are very similar to the previous version of the manuscript. Very small differences appears for the correlation results from Figures 6, 7 and 8. The RDA for

lacustrine samples also slightly changed (Figure 5E). The results do not change for the MANOVA results (Table 3), the 9 and 11 and the equations of the salinity-classes MBT-MAAT calibrations.

We do not found any other problems on the sample type classification of our dataset (e.g., no plant tissues or bones). Finally, lacustrine samples from the Lake Baikal are not any more included in the dataset.

**L119–132:** This section would benefit from some supplementary figures showing the environmental variables you are discussing, either as maps or histograms. As it stands, this is a pretty dense wall of text to get through without any visualization to help the reader understand what is being communicated.

Actually this section is already supported by several climographs extracted from sites within the ACA and available in Fig. S.1. We added this introducing sentence to guide the reader toward graphical representations of the bioclimatic variables from the ACA (l. 124-125): *"Temperature and precipitation maps for the ACA are provided in Fig. 2 from Dugerdil et al. (2025), and additional ACA climographs are shown in Fig. S.1.".*

**L142–143:** "The chemical characteristics of the ACA surface samples include pH and Electro-Conductivity (EC, Fig. 2, step 2), both measured ex-situ in the laboratory, even for lacustrine samples." Was there anything done to make sure that microbial respiration post sample collection did not cause the samples to become more acidic during transport? For the lacustrine samples, what is the logic that the pH or salinity of the pore water in the sediments is equivalent to the properties of the lake water? This is particularly concerning for pH, since the pH of sediment pore waters can change substantially (and therefore become different from the overlying water column) in even the upper few cm of the sediments (e.g., Bachmann et al., 2001; https://doi.org/10.1016/S0375-6742(01)00189-3).

We want to thanks you for these accurate concerns. We actually try to limit as much as possible the potential biases inherent to *ex-situ* measurement, mainly by storing the samples in freezer and conducting the analyses as soon as possible. We detail this statement in line 149 to 151 *"Following the field campaign, samples were stored at freezing temperatures and analysed at the earliest opportunity to minimise post-sampling alterations of brGDGT distributions and chemical properties that may arise from ongoing microbial metabolic activity.".*

Then, we discuss your concerns as potential limitations of the results of the present study. For instance, in lines 635 to 637: *"Moreover, the present approach is further limited by the ex-situ measurement of salinity and pH (Fig. 2). Ongoing microbial metabolic activity during sample transport and storage may have altered in-situ pH conditions prior to laboratory analysis, potentially biasing measured values.".*

About the lacustrine samples, we only work, for the new data, with lacustrine core tops, for brGDGT analysis as well as salinity and pH. Then, the values all come from sediment. We do not used pH and salinity measurement from the water column.

**3.1.1 brGDGT concentrations (L262–269):** Comparing concentration measurements across studies is really tricky because there is no authentic brGDGT standard. The C46 internal standard (Huguet et al., 2006) is useful for considering relative differences in concentration between samples run at the same time, but this method does not control for instrument drift (see Figure 4 of Huguet et al., 2006). Some work needs to be done to show whether the uncertainties introduced by this

shortcoming of the $C_{46}$ method pose a problem for the analysis here. Or, at the very least, some text needs to be added to acknowledge this shortcoming of the concentration data. The discussion of the concentration data here feels somewhat random since they do not appear to be discussed later in the text. If they are going to be brought up, it would be interesting to understand if concentration is related to any of the indices that describe the brGDGT distributions. For example, we found a relationship between brGDGT concentration and IR6Me in lacustrine sediment samples from Lake Baikal (Novak et al., 2025).

*This constraint regarding the estimation of absolute concentration is entirely accurate. Actually, the concentration data and cross-comparison are derived solely from the samples analysed in our laboratory in Lyon. This section utilizes a dataset comprising solely 234 samples from Dugerdil et al. (2021a), Cromartie et al. (2025) and Dugerdil et al. (2025). To clarify, we included the following text at the beginning of the section (l. 275-278): "The brGDGT absolute concentrations are estimated using the $C_{46}$ internal standard method. To avoid biases from instrument drift (Huguet et al., 2006), only the samples analysed in the geochemistry laboratory LGLTPE at ENS de Lyon are described here. The dataset comprises 234 samples from Armenia, Azerbaijan, China, Mongolia, Tajikistan, Russia and Uzbekistan, published in Dugerdil et al. (2021a), Cromartie et al. (2025) and Dugerdil et al. (2025).".*

**4.2.4 Toward a scale of confounding factor strength (L582–609):** This section does not present any sort of quantitation of the strength of the confounding factors in the temperature-methylation relationship, so the name of this subsection is not appropriate. Rather, this text is a discussion of previous work on the topic. Is there maybe a way to tie this discussion in with the previous sections a bit more coherently? As it stands, I struggled to understand what the takeaway message of these two paragraphs is.

*We agree that this section need a thorough reorganisation to better fit its title and its main goal. The idea here, is to discuss the relative importance of the environmental variables on the brGDGT distribution and its consequence on the reliability of the brGDGT indices as past proxies. In this section, we suggest that the ranking of controlling factor at the global scale is not similar in the ACA, and in other regional studies. To get there, we included in this section some results from the present study in order to suggest a ranking of variable importance for the ACA. This scale is also compared to previously published ranking from global and regional studies. The section is currently as follow (l. 616-645) "We have shown that brGDGT—environment relationships are influenced by several biases linked to confounding factors (Sect. 4.1; Figs. 5, 6, 7 and 8). In addition, interactions among these factors further limit proxy applicability (Sect. 4.2). A key next step to improve the reliability of brGDGTs as past environmental proxies is to rank the relative importance of these controlling influences. Although major environmental controlling factors have been identified globally (Naafs et al., 2017a; Dearing Crampton-Flood et al., 2020; Raberg et al., 2022a), their dominance hierarchy appears to vary by study region and archive.*

*At the global scale, temperature is consistently the primary control, followed by pH, regardless of sample type (Raberg et al., 2022a). This general ranking (temperature as primary factor and pH as secondary one) holds across environments, but sample type still modifies regression parameters: slopes and intercepts differ between soils, peats, and lake sediments (Naafs et al., 2017a, b). Then, additional influences depend on specific sample type. For example, seasonal effects are strongest in lake-based calibrations (Dang et al., 2018; Martínez-Sosa et al., 2021; Raberg et al., 2021, 2022a). Seasonality also affects soil calibrations (Deng et al., 2016; Dearing Crampton-Flood et al., 2020),*

*but it has been shown that the monthly temperature fluctuations reported by brGDGT indices were similar to the average temperature (Cao et al., 2018), mainly due to the slow turnover of brGDGT production and deposit in soil samples (Weijers et al., 2011). Overall, current evidence indicates that temperature and pH dominate brGDGT distributions, followed by sample type and proxy-specific modifiers such as seasonality, salinity, and aridity.*

*Our ACADB data suggest a different ranking of controlling factors for Arid Central Asia, and drylands more generally. Based on the MANOVA results (Table 3), salinity emerges as the dominant factor after temperature, followed by sample type, pH, and aridity, all contributing significantly to brGDGT variance. However, this ranking is derived from a limited subset of samples because salinity and pH measurements do not cover the entire dataset (Fig. 5). Moreover, the present approach is further limited by the ex-situ measurement of salinity and pH (Fig. 2). Ongoing microbial metabolic activity during sample transport and storage may have altered in-situ pH conditions prior to laboratory analysis, potentially biasing measured values. Nevertheless, the ACA ranking clearly differs from the global hierarchy. Previous regional and local studies have likewise reported differences from the global-scale rankings. In Chinese soils, pH is the primary control on brGDGT distributions, surpassing temperature (Wang et al., 2019). In hyper- to semi-arid settings, annual precipitation and aridity become more influential than either pH or temperature (Duan et al., 2022). For lacustrine archives on the Tibetan Plateau, salinity is the leading controlling factor (Liang et al., 2024). Such site-specific variability in factor importance supports the idea that community shifts, rather than physiological plasticity, primarily govern brGDGT responses to environmental parameters (Guo et al., 2021). This further underscores the need to evaluate controlling factors at local to regional scales when assessing the applicability of brGDGT-based indices.".*

**4.3.3 Confounding factors effect on temperature calibrations (L646–660):** Text needs to be added to the methods that explains how lakes were classified into the different "salinity classes." Specifically, the cutoff values for the different classes need to be given and, if they are not taken from another publication, justification for these cutoffs should also be provided.

We agree that this methodological point needed clarification. The cut-off values were already provided in the Supplementary Table S.3 but we forgot to add a cross-reference to this table in the manuscript. We added in the Method section, l. 165-167 "*The TDS values were used to provide four salinity classes (fresh, hyposaline, saline and hypersaline). The cut-off values were derived from Rusydi et al. (2018) and refined for the ACADB using a sensitivity analysis (Table S.3).*". We also add a reminder in the discussion with (l. 695-696) "*Salinity classes were based on a classification adapted from Rusydi et al. (2018). The TDS cut-offs were refined using sensitivity analysis and are reported in Table S.3.*".

In the current version of the manuscript, we made the difference between *thresholds* to describe different environmental response of brGDGT in front of continuous environmental variable and *cut-offs* for the class limits, in order to improve the understanding of the salinity and aridity classification.

**Figure 10:** This figure design is really confusing to me. What is going on with the z-scores at the top of the figure? What do the boxes with lines represent?

It is true that this figure is not very easy to follow. Especially because it presents statistic tests that are not commonly used in brGDGT calibration studies. The *z* provided on the upper part of the

figure are not *z*-scores but z-statistics produced by a statistical test. This test ensures that both the regression slope (*a*) and intercept (*b*) for each class is significantly different one another. The graphical links from the figure highlight the two groups tested. It is a simpler way to present the two groups tested. For instance, the *z*-statistic assessing the slope and intercept difference across the Fresh water and Hyposaline linear regression is z(a) = 2.7***, z(b) = 1.3.

To clarify this figure, we added panel names and titles to make sure the reader can understand that the *z* is a *z*-statistic and not a *z*-score. We also extend the graphical connector between the two tested groups with dashed lines. Finally, we refine the figure caption by "*Effect of salinity (i.e., the TDS of each surface sample) on the linear relationship between temperature (here MAAT) and the MBT'$_{5Me}$. Panel **A** shows the results of the z-statistic tests assessing significant differences in slopes and intercepts between salinity classes. Panel **B** presents linear regressions for fresh, hyposaline, saline, and hypersaline classes, based on the full ACADB. For each salinity group, temperature calibrations of the form MAAT = a x MBT'$_{5Me}$ + b are proposed (**B**). The z-statistic method (Clogg et al., 1995) evaluates the significance of slope (a) and intercept (b) differences across two classes (grey lines link the two classes with their z-statistics on Panel **A**). The p-values of the z-statistic are displayed with *** for p < 0.01, ** for p < 0.05, and * for p < 0.1.*".

**Equation 6:** This should probably be broken up into sub-equations (i.e., 6.1, 6.2, etc.) since there are multiple equations listed here. Also, five regressions are shown in Figure 10, but only three equations are given here. Why? This should be justified in the text or the other two equations provided.

Thank you one more time for the advice. We divided the equation section (6) into different sub-equations. We also added the all five equations of the linear models from Figure 10 (currently it is Figure 11). In the previous version of the manuscript, we decided to only show the equations for the three best $R^2$, but after tested the significance of each of them, it appears that all have a *p*-value below 0.001. We added this information in l. 705-706 "*All the linear models in Eq. (6) have a p-value < 0.001 when tested using 20,000-permutation significance tests.*".

**Minor Comments**

**L26:** I do not think we can really call brGDGTs a "new" proxy anymore.

True. Corrected.

**L29:** "…well preserved in lake sediments…" current wording is not grammatical.

The text has been modified.

**L30–33:** This sentence is written in a confusing way that is not understandable to someone who is not familiar with the brGDGT proxy.

This is true. We simplified the idea by spiting this sentence into two sentences (l. 29-34) "*The number of methyl groups on the aliphatic chain of brGDGT varies with ambient temperature, as demonstrated by Weijers et al., (2007) and De Jonge et al., (2014a), allowing their application as past temperature proxy. The correlation between temperature and the brGDGT degree of methylation is evident and linear, as demonstrated by both soil surface samples (De Jonge et al., 2014a; Dearing Crampton-Flood et al., 2020) and lacustrine surface sediments (Sun et al., 2011). A similar relationship is observed from laboratory experiments and simulations (Naafs et al., 2021; Halamka et al., 2023).*".

**L33:** "The relationship is clear and linear." I think there are missing words here. It is not clear what relationship you are talking about.

See response above.

**L34:** This sentence would benefit from consolidation.

The prior version indeed over-loaded. We have now (l. 34-37) *"The applications of brGDGT-based palaeothermometers encompass a diverse range of environments and archives, including tropical to Arctic lakes (Perez-Angel et al., 2020; Haggi et al., 2023; Raberg et al., 2022), acidic to alkaline lakes (Dang et al., 2016; Yang et al., 2014), freshwater to saline lakes (Dugerdil et al., 2021b; Wang et al., 2021; Robles et al., 2022; So et al., 2023), and sediments from loess–palaeosol sequences (Lin et al., 2024)."*.

**L35:** "acidic lakes"

Done.

**L38:** This statement could use a citation or two as an example.

This is true. We added some references to Duan et al. (2020, 2022) and Chen et al. (2021) for the impact of pH and precipitation on the brGDGT distribution.

**L55:** "pentamethyls" is a bit informal. Here and elsewhere this language should be replaced with "pentamethylated brGDGTs," adapted as appropriate for the molecules discussed in each instance.

The text has been modified here and to all occurrences of the words *penta-, tetra-* and *hexamethyls*.

**L60:** "This leads to a different temperature relationship between brGDGTs in soils and MAAT…"

Done.

**L68–69:** "…low organic matter content…"

Done.

**L85:** "Salinity is thought to influence…" is probably more appropriate since there is not (at least to my knowledge) a biochemical explanation for why the salinity would cause preferential synthesis of different brGDGTs.

Thank you, this wording is more nuanced.

**L89:** isn't MLR the typical abbreviation for multiple linear regression?

Actually, it is more logical but the MR abbreviation appears in numerous brGDGT studies such as De Jonge et al. (2014a), Chen et al. (2021), Li et al. (2017), Pérez-Angel et al. (2020).

**L95:** "…totaling 761 samples."

The text has been modified.

**L97:** What specific temperature variable(s) are you correlating to?

Done (l. 99-102) *"The present study tests this hypothesis by analysing modern brGDGT distributions against key climate parameters, including aridity, temperature (both Mean Annual Air Temperature, MAAT, and the seasonal mean temperature of Months Above Freezing, MAF), and*

*precipitation, as well as chemical characteristics such as pH, salinity, and sample type (soil or lacustrine).".*

**L112:** What does "the data location was randomly selected" mean?

It means that we randomly choose the coordinates of the sampling sites within the Azerbaijan borders. This approach is important in ecological sampling, to ensure that the sampling strategy is more representative of the ecological gradient. When sampling locations are chosen by the analysis on the field, the sampling is more likely uneven (e.g., on the field researchers may prefer select clear and known environments rather than ecotone or under-studied environments). We explain this point better in the manuscript with (l. 116-118): *"Prior to the field campaign, sample locations were defined using a randomized selection procedure within a GIS framework to enhance the bioclimatic and ecological representativeness of the dataset (Bunting et al., 2013).".*

**L138:** core tops rather than "top cores"

The text has been modified here and to all other occurrences.

**L193–197:** The justification for not looking at DC or DC' does not make a lot of sense to me. Should not all of the indices be considered?

The paper is already full of indices and statistical analysis. We therefore really have to make a choice. Moreover, the majority of the ACA studies are using CBT-derived indices, mainly by habits. DC may probably be more interesting, but we choose to postpone its analysis for further studies, and to focus first on already settled and well-known indices.

**L256:** "the size of the data is high" this reads a bit awkwardly. How about, "since the number of samples is large."

True. Corrected.

**2.5 Database compilations:** If you are going to include Lake Baikal data, the database is missing recently published surface sediment samples from Lake Baikal (Novak et al., 2025).

This dataset may not be part of this study (see response to major concerns above *Figure 1 and database concerns*). The data from the Lake Baikal are not any more included in the ACADB.

**Figure 3:** Some text is needed in the figure caption discussing how the bounds of the box and whisker plots are defined.

That point is truly important. The Figure 3 caption has been detailed with *"The hinges of the boxplots reprsent the 25th percentile (Q1) and 75th percentile (Q3), the central horizontal line indicates the median, and the whiskers extend to the most extreme data points that lie within 1.5 x the inter-quartile range (i.e., Q3 − Q1). Data points exceeding the whiskers are classified as outliers and are omitted from this representation. The compounds with the lowest abundances (mean value below 5%) are highlighted in panels **A1, A2, B1** and **B2**.".*

**3.1.2 brGDGT fractional abundances:** More figure callouts are needed in this section. Figure 3 is rather complicated, so it would be helpful to guide the reader through each of the panels.

This is true. The paragraph has been modified accordingly (l. 286-291) *"The brGDGT distribution is described from soil (Fig. 3A) and lacustrine samples (Fig. 3B). FAs with the lowest abundances (i.e., mean values below 5%) are presented in panels A1-B2. Concerning FAs in soil samples, the*

*prevalent compounds are IIa' (mean value ca. 30%), IIIa' (ca. 22%), Ia (ca. 14%), IIa (ca. 10%) and IIIa (ca. 6%, Fig. 3A). In lacustrine samples, the distribution is dominated by IIIa' (ca. 19%), IIa' (ca. 16%), Ia (ca. 12%), IIa (ca. 10%) and IIIa (ca. 9%, Fig. 3B). In contrast, compounds such as IIIb, IIIb', IIIb'', IIIc, IIIc', IIb'', IIc, and IIc' are rare in both soil and lacustrine samples, with average abundances ranging from 1% to 2% (Fig. 3A1-B2).".*

**L276:** "The 7-methyl isomers are more abundant in lacustrine…"

Done.

**L277:** If you are talking about Figure 3, these are box and whisker plots, not histograms.

True, thank you !

**L256–282:** The language here needs to be tightened up. When talking about the data, "trends" would imply some sort of regression exercise that generated an estimate of slope. Here, it is more important to talk about differences in the median values in the sample sets separated by aridity class (which is what the box and whisker plots are showing). This could be coupled with significance tests in differences of the mean.

Yes that is definitely true. We decided to reformulate this paragraph without adding variance analyses such as MANOVA and Tukey tests here, in order to not overload this section, only aiming to describe the data distribution and the primary observation that the aridity may be a controlling factor of the brGDGT distribution. The new version of the text (l. 293-300) is as follow: "*For both sample types, the boxplots reveal that brGDGT distributions vary across the different aridity classes, with median FAs shifting toward higher values with drier aridity classes: median IIIa increases in the humid class, while IIIa' and IIIa'' are higher in the arid and hyper-arid classes. A similar median shift is observed between IIc and IIc'. Additionally, IIa increases with wetter aridity classes, while the IIa' distribution remains largely insensitive to changes in aridity classes for lacustrine but decrease with higher humidity for soil samples. Compounds Ia, Ib and Ic exhibit discernible variations between aridity classes, although the observed shifts are not unequivocal. Finally, aridity control is less evident in other low-abundance compounds, including the IIIb, IIIc, and IIb, and all 7-methyl isomers.*".

**L292–293:** What do you mean by the aridity effect? This is not clear from the text.

We mean that for soil, the sample distribution along the ternary plots is different across the aridity classes. And this observation is mitigated for lacustrine samples. The sentence has been refined with (l. 310-311) "*For lacustrine samples, the distribution across aridity classes appears less contrasted than for soil samples.*".

**L304:** I would not say this is a "clear aridity gradient" since there is substantial overlap between the distributions plotted on the upper axes of the plots in Figure 5. This result is, perhaps, suggestive of an aridity gradient.

This is true that the data do not support a clear statement. We slightly nuanced the sentence by (l. 321-326) "*About the distribution of aridity classes over this multivariate space (highlighted by the upper and lateral sample densities in Fig. 5A and B), the data distribution suggest that an aridity gradient can be superimposed on the isomer gradient (...)*".

**3.2.1:** Some text explaining why this analysis was done and what the VIF value means would be really useful to guide the reader through this dense section of the text.

The justification of the choice of statistical analyses is already presented in the Method section. However, a reminder is indeed useful here, especially because we present the RDA and VIF analyses from different dataset (climate data and brGDGT/climate data). Then, we add an introductory sentence in l. 334-339 as *"The selection of bioclimatic parameters that can be reliably reconstructed from fossil proxies is essential (Salonen et al., 2019). To evaluate this, we conducted (1) a multivariate analysis on worldclim2.1 data extracted at the ACADB sampling locations, to identify the primary and secondary bioclimatic gradients and their main parameter contributors, and (2) a Variance Inflation Factor (VIF) analysis to quantify multicollinearity within the brGDGT-bioclimate multi-variate space of the ACADB. The goal is to confirm which bioclimatic parameters are the most informative and ensure they are as statistically independent as possible, in order to minimize biased climate reconstructions."*.

**L341–345:** Figure callouts are needed here.

True, we add cross-references to Figure 5D and E to clarify when we consider soil and lacustrine samples in this section.

**L364:** The meaning of the "*" needs to be explained, or the p-value can just be written as p << 0.001 or however else is appropriate for each relationship.

The meaning of the "*" is given in caption of the Figure 6 which is described in this paragraph. However, it is better to give in the plain text, their meaning here. Done with (l. 387-388) *"The $R^2$ (\*\*\* indicates p-value < 0.001) are given for each relationship and for both subsets of sample type (i.e., soil and lacustrine samples)."*.

**L361–372:** Is there a figure associated with these tests and statements (presumably Figure 6)? Callouts are needed.

Yes, it is associated to the Figure 6. The first call-out were already present on line 362. We add one call-out later on l. 394 with *"Comparing the ACADB to the WDB in Fig. 6"*.

**L372:** Is there a figure or statistical test (preferably both) you can point to that indicates that "hyper-arid samples often have extreme values" ?

This sentence was only based on graphical observations from Fig. 6, then, since we did not perform a particular test to validate this statement, we prefer remove it on the revised version of the manuscript.

**L384–386:** I think you mean "…there is a strong correlation between…"

No exactly, we mean that the CBT'-IR$_{6Me}$ and CBT'-IR'$_{6+7Me}$ relationships are more similar in term of slope and correlation for soil than lacustrine samples. This idea has been refined as follow (l. 410-412): *"Finally, the CBT'-IR$_{6Me}$ and CBT'-IR'$_{6+7Me}$ relationships are more similar in term of slope and correlation for soil than lacustrine samples ($R^2$ of 0.87\*\*\* and 0.80\*\*\* for soil, 0.80\*\*\* and 0.13\*\*\* for lacustrine, respectively in Fig. 7G and H)."*.

**L428–429:** The confounding factor in relation to what? Estimating MAAT from one of the methylation indices?

Yes exactly, as we generally consider the brGDGT to be a proxy of MAAT. The beginning of this paragraph has been modified as follow (l. 454-460) *"Comparing the ACADB results with other studies, the question of the relative importance of the confounding factor on brGDGT*

*palaeothermometer applications from drylands is raised. The effects of the confounding factors on the past temperature estimation (…)"*.

**L454:** "attenuation" is not the right word to use. Weakening is probably what you mean.

True. Corrected.

**L475:** typo here.

True. Corrected.

**L500:** Again, a confounding factor for what?

In this case, we agree that it is more relevant to defined the salinity as a controlling factor than a confounding factor as we are focusing on the relationships between salinity and isomer ratios. The paragraph has been modified accordingly.

**L512–514:** "These thresholds of salinity classes are attenuated…" I do not really understand what you mean here. What thresholds do you mean? I do not see any in the Figure that is referenced. I understand that you are talking about the difference in the correlation strength to salinity of IR6Me vs. IR6+7Me, but the language here is hard to follow.

This is true. We detected these thresholds by sensitivity analysis, now added as supplementary figure S.6. We therefore modified the text following (l. 538-544): *"IR$_{6Me}$ is well correlated with salinity for low salinity values, particularly in lacustrine samples (i.e., mainly fresh-water lakes, Fig. 8A and Wang et al. 2021b), while IR$_{7Me}$ is more significant for higher salinity ranges (Fig. 8B). Below TDS values of ca. 1,000 mg.L$^{-1}$, the IR$_6$Me-salinity relationship shows lower data dispersion that above this threshold (Fig. S.6A). For the IR$_{7Me}$ index, a saturation effect due to low 7-methyls isomer mitigate the IR$_{7Me}$-salinity relationship, below TDS values of ca. 11,000 mg.L$^{-1}$ (Fig. S.6B). The different IR response to salinity below and above these salinity thresholds (TDS in [1,000; 11,000] mg.L$^{-1}$ in Fig. 8A and B) is attenuated when both 6- and 7-methyls are included in the ratio over 5-methyls (i.e., IR$_{6+7Me}$ and IR'$_{6+7Me}$ indices, Fig. 8C and D)."*.

**L526–527:** What is a "correlation disruption"?

The wording was in fact not appropriate. We refine the sentence by (l. 557-559): *"Particularly, panels (A) and (B) from Fig. 7, show that the cyclisation indices exhibit a piecewise, dual-slope response across a threshold, indicating a non-linear, threshold-dependent relationship rather than a single unified pH control."*.

**L528:** "global Chinese soils" seems self-contradictory.

Here we want to emphasize the fact that this dataset is from the whole Chinese area, including both samples from arid and humid environments. The sentence has been refined as follow (l. 560-561): *"At the Chinese soils scale, including samples from arid and humid environments, (…)"*.

**L562:** What is a "not recalibrated temperature reconstruction" ?

Here we just want to insist on the classical linear calibration for brGDGT-based temperature reconstruction. However, it sounds useless. The sentence has been refined as (l. 594-595) *"The salinity effect on temperature reconstruction may result in a temperature over estimation of more than 2°C (Liang et al., 2024)"*.

**L564:** "over-representation" Also, not really sure what you mean here.

True, it is better to use the same wording that in previous sections. This *"over-representation"* has been replaced by *"average fractional abundance"*.

**L615:** "constrained" is not the correct word to use here. I think you mean "led" but you should make sure of that.

True, corrected.

**L643:** What is a "locally recalibrated global calibration" ?

Here it just means, regional calibration. The sentence was changed to (l. 685-686) *"Using a local or regional calibration (Dugerdil et al., 2021a; Chen et al., 2021), generally reduces this offset.".*

**L644-645:** Could some of this not be because of the typically different seasonality of brGDGT and pollen temperatures? As in, brGDGTs typically are a proxy of MAAT or MAF while pollen is typically mean temperature of the warmest month or mean temperature of the coldest month? Temperature change is expected to have a different magnitude across different seasons from climate model simulations and modern observations (Feldl and Merlis, 2021, is an example from the high latitudes).

Actually, whatever the climate parameter reconstructed, the pollen-based climate reconstructions always have smaller temperature variation reconstructed along the Holocene (Dugerdil et al., 2021b). I guess here, it is more because of the different statistics and models used to reconstruct temperature from pollen and brGDGTs rather than a real seasonal effect. Mainly, the simple linear relationships commonly used for brGDGT reconstructions are more sensitive than multi-variate and nonlinear space-for-time approaches used in pollen studies.

**L664–665:** I do not think "lead" is communicating your intended meaning in this sentence, but I cannot figure out what you are trying to say so I cannot suggest a better word.

True. Maybe "develop" or "train" is a better word. We rephrase the whole sentence by (l. 714-715) *"Developing calibrations for specific sediment types can help reduce biases in brGDGT-based climate reconstructions, but some challenges remain.".*

**L695:** Rather than "mitigated" I think you mean "limited."

True. Corrected.

---

## Referee Report (RR1)

**Review #2 of Dugerdil *et al.* for *Biogeosciences***

**Joseph B. Novak**

**Summary**

I thank the authors' for their efforts to address my previous comments. After rereading the manuscript, I found a handful of issues that need to be addressed, the most critical of which is my comment regarding line 567. Otherwise, I congratulate the authors on presenting a thorough manuscript that advances our understanding of branched GDGT molecules in complex arid environments.

Warm regards,

Joseph B. Novak

**Major Comments**

**Figure 1 and the soils dataset:** some sort of justification should be given for the inclusion of the moss polsters in the soils dataset. Strictly speaking, a moss polster is not a type of soil, and I foresee this as a sample type that many who primarily work on the application of the brGDGT proxy to geologic deposits will not be familiar with. It is probably also best to mark these samples with a different marker face color so that they can be easily picked out on the map. Also, the color scheme for this figure is not colorblind accessible, particularly panel D (red symbols with green background). The simplest solution to that would be to change the marker face color of the red symbols to something else.

**L567:** It looks like a new sentence (or perhaps paragraph) was started but not finished here.

**Minor Comments**

**L14:** I think "Despite this" is a bit confusing since it seems reasonable to expect a weak relationship between brGDGT methylation and environmental temperature given the significance relationship between the various aspects of the brGDGT distributions and salinity, pH, etc.

**L24:** I think "paleotemperatures" is probably a better term here

**L31–33:** It might be best to specific that you are talking about the global dataset here since you highlight in the abstract that there is a weak relationship between brGDGT methylation and temperature in ACA.

**L43–46:** "These two important indices" follows a sentence where three indices are defined. Probably best to specific which two indices you mean, or to break the previous sentence into two sentences.

**L51:** "have" rather than "has"

**L63:** the first "they" in this sentence should be replaced with a noun, as it reads it is a bit confusing whether you mean the brGDGTs or the environmental variables.

**L76:** What is meant by "specific isomer distribution?" This is a bit confusing – do you mean that the distribution of 5-methyl vs. 6-methyl vs. 7-methyl isomers is more variable?

**L82:** Some additional citations are appropriate here to further support the idea that bacteria community composition complicates application of the brGDGT paleothermometer. I suggest:

(Ajallooeian et al., 2025) https://doi.org/10.1029/2025JG009132

(De Jonge et al., 2019) https://doi.org/10.1016/j.orggeochem.2019.07.006

**L84:** Probably best to add "in some environmental contexts" since substantial variations in $IR_{6Me}$ have been seen in freshwater environments also (e.g., Novak et al., 2025).

**Section 2.3:** Would it be appropriate to add a citation to the seminal paper by Hopmans et al. (2016) since the method used here separates the 5-methyl and 6-methyl brGDGT isomers?

**L218:** I think you mean Figure 1D

**L283:** *solonchak* should be defined since this is an English language journal. It can be as simple as "*solonchak* (salt marsh) samples."

**L593–594:** By "actual observations," do you mean measurements of lake surface salinity? Also, it would be best to specific that So et al. study referenced here is not from the ACA region (which is fine, the data are obviously relevant, but the distinction should be made).

**Figure 10B:** by "over cold" and "over warm" do you mean cold and warm biased?

**4.4 section title and subsection titles:** I suggest changing "in the past" to "in the geologic record" since I think this better describes what you are discussing here

**L737:** check subscript here

---

## Author Response (AR2)

**Review #2 of Dugerdil et al. for Biogeosciences Joseph B. Novak**

**Summary**

I thank the authors' for their efforts to address my previous comments. After rereading the manuscript, I found a handful of issues that need to be addressed, the most critical of which is my comment regarding line 567. Otherwise, I congratulate the authors on presenting a thorough manuscript that advances our understanding of branched GDGT molecules in complex arid environments.
Warm regards,
Joseph B. Novak

**Major Comments**

**Figure 1 and the soils dataset**: some sort of justification should be given for the inclusion of the moss polsters in the soils dataset. Strictly speaking, a moss polster is not a type of soil, and I foresee this as a sample type that many who primarily work on the application of the brGDGT proxy to geologic deposits will not be familiar with. It is probably also best to mark these samples with a different marker face color so that they can be easily picked out on the map. Also, the color scheme for this figure is not colorblind accessible, particularly panel D (red symbols with green background). The simplest solution to that would be to change the marker face color of the red symbols to something else.
I agree with you that this point has to be clarified. Actually, only few samples from the NMSDB (Mongolia and Siberia) are moss polster. We detailed this point with " *The NMSDB is composed by 27 moss polsters, 15 soil samples and two lacustrine core tops. As the GDGT distribution is similar for soil samples and moss polster (Dugerdil et al., 2021a), we included the moss polsters within the soil sample type in the present study*". For colorblind accessibility, we changed the red color into clear orange. After testing this new color chat with colorblind website, it appears to be better than the previous one.

**L567**: It looks like a new sentence (or perhaps paragraph) was started but not finished here.
I am very sorry for this forget. It was actually a work-in-process sentence to conclude the modification of we made to answer some of your conserns from the previous review step. The idea was to clearly settle which index type (isomer or cyclisation ratios) are the most suitable to track pH change from our dataset. We removed the sentence L567 and changed it by "*From the ACADB, isomer ratios outperforms cyclisation indices to track pH variations, as isomer ratios follow unimodal relationships with pH, while cyclization indices follow dual-slope relationships.*".

**Minor Comments**

**L14**: I think "Despite this" is a bit confusing since it seems reasonable to expect a weak relationship between brGDGT methylation and environmental temperature given the significance relationship between the various aspects of the brGDGT distributions and salinity, pH, etc.
That is true, we changed it into "Thus".

**L24**: I think "paleotemperatures" is probably a better term here
Yes, we agree that it makes the sentence more digest.

**L31–33**: It might be best to specific that you are talking about the global dataset here since you highlight in the abstract that there is a weak relationship between brGDGT methylation and temperature in ACA.

True, we gave more details about the way to obtain this result by changing the sentence into: *"Analyses of worldwide calibration datasets indicate that variations of the number of methyl groups on the brGDGT aliphatic chains are primarily controlled by ambient temperature, enabling their application as proxies for past temperature (Weijers et al., 2007; De Jonge et al., 2014)."*.

**L43–46**: "These two important indices" follows a sentence where three indices are defined. Probably best to specific which two indices you mean, or to break the previous sentence into two sentences.
Thank you for this nice reading, we change the sentence with *"$MBT'_{5Me}$ is now widely adopted to calibrate the reconstruction of MAAT in the past by linear relationships, while $CBT_{5Me}$ and IR are used to infer past pH variations"*.

**L51**: "have" rather than "has"
Corrected.

**L63**: the first "they" in this sentence should be replaced with a noun, as it reads it is a bit confusing whether you mean the brGDGTs or the environmental variables.
Changed for *"these controlling factors"*.

**L76**: What is meant by "specific isomer distribution?" This is a bit confusing – do you mean that the distribution of 5-methyl vs. 6-methyl vs. 7-methyl isomers is more variable?
I guess *"a particular isomer distribution"* is a better term than *"specific isomer distribution"* as we want to introduce the fact that the 6- and 7- methyl abundances over the 5-methyl's one is more important in dryland than in other context.

**L82**: Some additional citations are appropriate here to further support the idea that bacteria community composition complicates application of the brGDGT paleothermometer. I suggest: (Ajallooeian et al., 2025) https://doi.org/10.1029/2025JG009132 (De Jonge et al., 2019) https://doi.org/10.1016/j.orggeochem.2019.07.006
Thank you for the references, we added them into the main text.

**L84**: Probably best to add "in some environmental contexts" since substantial variations in IR6Me have been seen in freshwater environments also (e.g., Novak et al., 2025).
This is also an important point to highlight that the relationships between salinity and the brGDGT distribution, particularly in terms of isomer ratio are not yet clear. We changed this paragraph with *"Salinity is thought to influence the relative number of 5-, 6- and 7-methyl isomers in some environmental contexts (Wang et al., 2021). Besides, substantial variation of the 5- over 6-methyl ratio ($IR_{6Me}$) are also observable in freshwater environments (e.g., Nowak et al., 2025). These variations of isomer abundances impact the $MBT'_{5Me}$- and $MBT''_{6Me}$-based temperature reconstructions (Kou et al., 2022; So et al., 2023)."*.

**Section 2.3**: Would it be appropriate to add a citation to the seminal paper by Hopmans et al. (2016) since the method used here separates the 5-methyl and 6-methyl brGDGT isomers?
Yes that is important too. We added *" This method allows to distinguish 5-methyl and 6-methyl isomers for each compound (Hopmans et al., 2016)."* in the method section after the presentation of the HPLC.

**L218**: I think you mean Figure 1D
Yes ! Thank you.

**L283**: solonchak should be defined since this is an English language journal. It can be as simple as "solonchak (salt marsh) samples."

Done.

**L593–594**: By "actual observations," do you mean measurements of lake surface salinity? Also, it would be best to specific that So et al. study referenced here is not from the ACA region (which is fine, the data are obviously relevant, but the distinction should be made).
The data are water column salinity measurements. We precised this information within the citation. For the citation of So et al. (2023), we changed it to "*(e.g., in the Great Salt Lake located in North American drylands, So et al., 2023).*".

**Figure 10B**: by "over cold" and "over warm" do you mean cold and warm biased?
4.4 section title and subsection titles: I suggest changing "in the past" to "in the geologic record" since I think this better describes what you are discussing here
To make this point clearer, we add two information in the caption. (1) " *The temperature biases are expressed with DeltaMAAT = MAAT$_{observed}$ - MAAT$_{predicted}$.*" and (2) "*Annotations Over cold and Over warm illustrate cold and warm biases, respectively.*".

**L737**: check subscript here
Nice catch !

**Editor comments**

Public justification (visible to the public if the article is accepted and published):
Dear authors,

Thank you for the first round of revision. The reviewer is satisfied with the edits made, but have few more comments/edits requests, the most critical comment pertains to L 567 (see their comments attached ).

While I sent this revision to reviewers for their evaluation, I also performed some review on the manuscript itself, and have the following comments/suggestions:
(note: the line numbers I mention below refers to you track changed document)

**L 4:** this study investigates
Done.

Somewhere around this line, it would be helpful if the authors elaborate on the rationale and research questions that need to be addressed. Highlighting what is really the difference and novelty that this study brings relative to the previously published paper.

**L 18-19**: I would revise it as "indirectly influence by multiple factors"
Done.

**L 35:** Similar relationship (instead of "same relationship")
Done.

**L69**: the interaction across potential contributing factors
Done.

**Data availability:** I appreciate the intent for open access, and it would be helpful to have the link available in the next revision as well. Also, to consider some of the comments about the type of samples from the reviewers, I encourage the authors to clearly indicate in their compiled dataset the type of samples (for their own and the other published).

Done. We add in the *Data availability* section the PANGAEA reference and link to access the data.

**Table 1:** Rephrase the second sentence as Samples with an asterisks (*) represent published data from Dugerdil et al. (2025)
The sentence has been rephrased as follow : "*Data from Dugerdil et al. (2025c) are highlighted by a * for a total sum of 162 surface samples*".

Please make sure that the supplementary files are revised accordingly.
The caption of the supplementary figure S1 has been changed in "*Geographical and biological presentation of the surface sites of the ACADB analysed in this study and previously published in Dugerdil et al. (2025c)*".

**Figure S1:** Are these climographs based on a single year or an average of a timespan? Please clarify (also if the data are instrumental datasets or reanalysis).
*The data are from the worldclim2.1 database, then there are average values over about 30 years. We added in the figure caption "Climate data represent averages over the period ca. 1970-2000 (Fick et Hijmans, 2017).".* We also added precision about the worldclim2.1 database with: "*The climate parameters were extracted from interpolated climate data from worldclim2.1 (Fick et Hijmans, 2017) at the sample location of the ACADB sites.*".

**Table S1:** please revise the caption to accurately reflect the status. These are not new samples, but already published, right?
Yes that is true. We modified the caption (see comment above).

For Table S2: please indicate the sample type as you did for Table S1.
Yes we add a column of the number of soil and lacustrine samples for each reference.

**Notification to the authors:**
Your "Short summary" text in the database (MS records) contains the abbreviation GDGT. Please provide at least one written-out version to make it better understandable for non-experts. Please remember that there is a character-limitation for the short summary text of max. 500 characters (including spaces).

---

## Author Response (AR3)

Dear authors,

Thank you for revising your manuscript. Thank you for including the dataset and code as well. I am happy to publish it. I only have one more comment at L609, do you mean "modern observations" instead of actual? (actual in your writing reads like "existing in fact" and it does not read parallel to "the Holocene". Lastly, please double check for potential symbol errors and typographical errors.

Thank you for your patience during the review process.
Best wishes,
Voary

Dear editor,
Thank you to point toward this error. It is very common as french speaker to make the confusion between actual and modern. We changed this throughout the manuscript.
Few typographical errors have also been corrected.
Thank you for your very careful reading and involvement in the editorial process for Biogeosciences.
Sincerely,
Lucas Dugerdil